# Rethinking Time-Series Imputation as Conditional Inference along Temporal Evolution

**Yu Fan** [1]   **Yang Yang** [1 2]   **Yufan Guo** [1]   **Huazhong Yang** [1]   **Pengjun Wang** [1]

## Abstract

Real-world time-series data often suffer from missing observations, hindering long-range temporal modeling. However, most existing imputation methods formulate imputation as conditional reconstruction over limited context, which restricts temporal information propagation and fails to explicitly model temporal evolution. To overcome this limitation, we propose the Conditional Temporal Inference Paradigm (CTIP), which formulates time-series imputation as conditional inference along temporal evolution. Under this paradigm, we introduce CBiT, which leverages a history compression mechanism to encode long-range history into a compact latent space for history-conditioned temporal imputation. In addition, we adopt a partitioned modeling strategy that distinguishes historical context and temporal imputation targets with only lightweight post-attention processing. Extensive experiments on multiple public benchmarks show that CBiT improves imputation accuracy by reducing Masked MAE and Masked RMSE by **27.3%** and **18.6%**, respectively, across different missing rates.

## 1. Introduction

In real-world time series, missing values are usually caused by sensor failures (Sharma et al., 2024) or irregular data collection, which fragment the temporal structure and hinder continuous state evolution modeling and downstream tasks. As a result, time-series imputation (Wang et al., 2024a) becomes increasingly important for maintaining temporal consistency.

Early time-series imputation methods were based on statistical assumptions (Yozgatligil et al., 2013; Moritz &

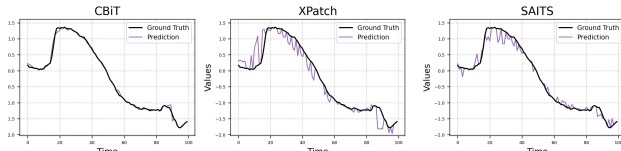

*Figure 1.* Visualization of imputation results on the Power dataset under a 50% missing rate. The black line denotes the ground truth, while the colored lines correspond to the imputed values produced by SAITS, XPatch, and the proposed CBiT model.

Bartz-Beielstein, 2017), including low-rank decomposition (Gillard & Usevich, 2018), but are limited in modeling nonlinear dynamics (Jung & Johansson, 2022) and complex data distributions (Yu et al., 2024). Recent advances in time-series imputation increasingly formulate missing-value recovery as structured conditional learning (Eskandari et al., 2025), where models learn latent representations from incomplete sequences (Ma et al., 2021). Related works further incorporate structural constraints (LIU et al., 2023), low-dimensional representations, and distribution-level learning objectives within unified end-to-end frameworks (Wang et al., 2025). Despite these advances, most existing time-series imputation methods (Ahmed et al., 2025; Yang et al., 2025a) implicitly adopt a local conditional modeling assumption (Lin et al., 2022), aggregating observations within limited temporal windows and focusing on local reconstruction consistency rather than the temporal evolution process (Bach et al., 2016), thus treating time as an indexing mechanism rather than a state-evolving dimension. Consequently, such models favor static correlations over genuine dynamic dependencies, weakening state transitions and cross-interval temporal relationships. Meanwhile, existing joint forecasting–imputation frameworks (Challu et al., 2022; Leppich et al., 2025) emphasize temporal modeling for prediction, whereas imputation largely retains local or reconstruction-based paradigms, with only indirect connections to temporal modeling.

Based on the above analysis, we propose the Conditional Temporal Inference Paradigm (CTIP), which reformulates time-series imputation as conditional inference over continuous temporal evolution. Under CTIP, missing values at different time points are inferred along a shared temporal

[1]Department of Electronic Engineering, Tsinghua University, Beijing 100084, China [2]Smartbow, Beijing 100096, China. Correspondence to: Pengjun Wang <wangpj@mail.tsinghua.edu.cn>.

*Proceedings of the 43rd International Conference on Machine Learning*, Seoul, South Korea. PMLR 306, 2026. Copyright 2026 by the author(s).

evolution process, which unifies local reconstruction and temporal evolution inference through a coherent conditional modeling structure. Figure 2 schematically contrasts the proposed paradigm with prior local reconstruction-based imputation methods in terms of temporal modeling.

When instantiated under long temporal histories, the proposed paradigm entails two key challenges. First, incorporating long historical contexts substantially increases computational cost during training and inference. Second, directly modeling historical information and imputation targets in a unified manner implicitly treats them as homogeneous signal sources, despite their fundamentally different roles: historical sequences provide long-term contextual and evolutionary cues, whereas imputation targets correspond to local, incomplete, and highly uncertain states. This mismatch can bias the model toward overly smooth global patterns and suppress informative but structurally weaker temporal dependencies.

To support the unified paradigm under long histories with reduced computational complexity, we introduce the Cross-Block Imputation Transformer (CBiT), which compresses historical states into compact latent representations and adopts a partitioned modeling strategy to separately capture long-term temporal evolution and local imputation uncertainty with reduced historical modeling cost. The imputation results are visualized in Figure 1.

Based on the above analysis, we summarize our contributions as follows:
(i) We propose the CTIP, which reframes time-series imputation as conditional inference over temporal evolution, bridging local reconstruction and temporal modeling.
(ii) Under this paradigm, we develop CBiT, an efficient time-series imputation model for long-history and high-dimensional time series that explicitly accounts for the heterogeneous roles of historical context and imputation targets, enabling stable and tractable inference.
(iii) Extensive experiments across multiple public time-series benchmarks, under varying missing patterns, missing rates, and supervision settings, demonstrate that the proposed method achieves superior performance in most cases in terms of imputation accuracy and stability.

## 2. Preliminaries

### 2.1. Transformer Multi-Head Attention

Multi-head attention (MHA) is a core component of Transformer (Vaswani et al., 2017), which models token interactions via multiple parallel attention heads. Specifically, given an input token set $\mathbf{Z} = \{\mathbf{z}_i\}_{i \in \mathcal{I}}$ with $\mathbf{z}_i \in \mathbb{R}^D$, the $h$-th attention head projects each token into query, key, and value vectors: $\mathbf{q}_i^{(h)} = \mathbf{W}_Q^{(h)}\mathbf{z}_i$, $\mathbf{k}_i^{(h)} =$

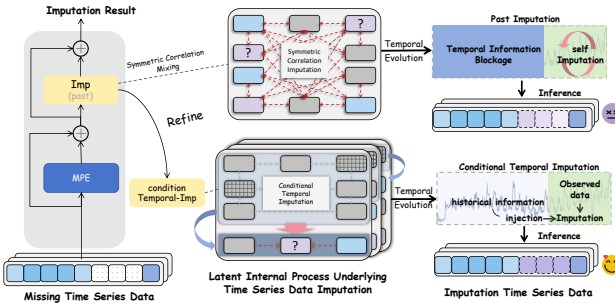

*Figure 2.* Conceptual illustration of the proposed CTIP paradigm in contrast to conventional local reconstruction-based imputation approaches.

$\mathbf{W}_K^{(h)}\mathbf{z}_i$, $\mathbf{v}_i^{(h)} = \mathbf{W}_V^{(h)}\mathbf{z}_i$, where $\mathbf{W}_Q^{(h)}, \mathbf{W}_K^{(h)}, \mathbf{W}_V^{(h)} \in \mathbb{R}^{D \times d}$, $d = D/H$, $H$ is the number of heads, and $\mathcal{I}$ indexes the input tokens.

The scaled dot-product attention weight from token $i$ to token $j$ in the $h$-th head is defined as

$$A_{i,j}^{(h)} = \frac{\exp\left(\langle \mathbf{q}_i^{(h)}, \mathbf{k}_j^{(h)} \rangle / \sqrt{d}\right)}{\sum_{j' \in \mathcal{I}} \exp\left(\langle \mathbf{q}_i^{(h)}, \mathbf{k}_{j'}^{(h)} \rangle / \sqrt{d}\right)}.$$

The output of the $h$-th attention head for token $i$ is then given by $\mathbf{o}_i^{(h)} = \sum_{j \in \mathcal{I}} A_{i,j}^{(h)} \mathbf{v}_j^{(h)}$.

The multi-head outputs are concatenated and linearly projected as $\mathbf{o}_i = \text{Concat}\left(\mathbf{o}_i^{(1)}, \ldots, \mathbf{o}_i^{(H)}\right)$, where $\text{Concat}(\cdot)$ denotes the concatenation of the outputs from all $H$ attention heads along the feature dimension. The resulting output sequence is given by $\mathbf{O} = \{\mathbf{o}_i\}_{i \in \mathcal{I}} \in \mathbb{R}^{|\mathcal{I}| \times (Hd)}$.

### 2.2. Perceiver-Style Latent Attention

Perceiver (Jaegle et al., 2021b) employs learnable latent tokens to compress high-dimensional inputs through cross-attention, avoiding full self-attention over all input tokens.

Formally, given an input token sequence $\mathcal{X} = \{\mathbf{x}_t\}_{t=1}^L$, and a set of $N$ learnable latent tokens $\mathcal{Z}^{(0)} = \{\mathbf{z}_n\}_{n=1}^N$ with $N \ll L$, latent representations are obtained via cross-attention:

$$\mathcal{Z} = \text{CrossAttn}\left(Q = \mathcal{Z}^{(0)}, \; K = \mathcal{X}, \; V = \mathcal{X}\right).$$

This operation compresses high-dimensional inputs into compact latents, reducing attention complexity from $\mathcal{O}(L^2)$ to $\mathcal{O}(NL)$ and enabling scalable inference.

### 2.3. Problem Formulation

**Definition 2.1** (Time Series). A multivariate time series is defined as a tuple $\mathcal{D} = (\mathbf{X}, \mathbf{M}, \mathcal{T})$, where $\mathbf{X} = \{x_t\}_{t=1}^{\mathcal{T}} \in \mathbb{R}^{\mathcal{T} \times C}$ denotes the observation matrix of a sequence with

length $\mathcal{T}$ and $C$ variables, $\mathbf{M} \in \{0,1\}^{\mathcal{T} \times C}$ is the missingness mask such that $m_{t,c} = 1$ indicates $x_{t,c}$ is missing and $m_{t,c} = 0$ indicates it is observed, and $\mathcal{T}$ denotes the temporal length of the sequence. The masked input representation is defined as

$$\mathbf{X}_{t,c} = \begin{cases} 0, & \text{if } m_{t,c} = 1, \\ x_{t,c}, & \text{if } m_{t,c} = 0. \end{cases}$$

**Definition 2.2** (Conditional Temporal Inference Paradigm). Given a multivariate time series $\mathcal{D} = (\mathbf{X}, \mathbf{M}, \mathcal{T})$ of length $\mathcal{T}$, the sequence is partitioned into temporal windows of length $T$ for imputation. Let $L$ denote the history length and $\Delta$ the imputation horizon, with $L + \Delta \leq T$. Each window is decomposed into a history segment $\mathbf{X}_{\text{hist}} = \{x_t\}_{t=1}^{L}$ and a target imputation segment $\mathbf{X}_{\text{imp}} = \{x_t\}_{t=L+1}^{L+\Delta}$, where $\mathbf{X}_{\text{imp}}^{(m=0)}$ and $\mathbf{X}_{\text{imp}}^{(m=1)}$ denote the observed and missing variables indicated by $\mathbf{M}$, respectively.

Under this paradigm, past-conditioned time-series imputation aims to estimate the missing variables via

$$\hat{\mathbf{X}}_{\text{imp}}^{(m=1)} = f_\theta \Big( \mathbf{X}_{\text{hist}}, \mathbf{X}_{\text{imp}}^{(m=0)}, \mathbf{M}, \Delta \Big),$$

$$f_\theta(\cdot) \sim p_\theta \Big( \mathbf{X}_{\text{imp}}^{(m=1)} \, \Big| \, \mathbf{X}_{\text{hist}}, \mathbf{X}_{\text{imp}}^{(m=0)}, \mathbf{M}, \Delta \Big),$$

where $f_\theta$ denotes a parameterized imputation function.

## 3. Theoretical Motivation

**Assumption 3.1** (Non-degenerate Temporal Information) We assume that time-series data exhibit non-degenerate global temporal information, with the overall temporal context not fully determined by local observations within a window.

$$p_\theta \Big( X_{\text{imp}}^{(m=1)} \, \Big| \, X_{\text{hist}}, X_{\text{imp}}^{(m=0)}, M, \Delta \Big) \neq$$
$$p_\theta \Big( X_{\text{imp}}^{(m=1)} \, \Big| \, X_{\text{imp}}^{(m=0)}, M, \Delta \Big). \quad (1)$$

In time-series imputation, the objective is to approximate the true conditional distribution $p_\theta \Big( X_{\text{imp}}^{(m=1)} \, \Big| \, \mathcal{I} \Big)$, where $\mathcal{I}$ denotes the available conditioning information. Under this formulation, ignoring historical context leads to a misspecified conditional distribution under a reduced information set, which cannot faithfully represent the imputation objective. In particular, without explicitly incorporating historical context, the objective degenerates to the local conditional distribution $p \Big( X_{\text{imp}}^{(m=1)} \, \Big| \, X_{\text{imp}}^{(m=0)}, M, \Delta \Big)$. By the law of total probability, this distribution admits the decomposition

$$p \Big( X_{\text{imp}}^{(m=1)} \, \Big| \, X_{\text{imp}}^{(m=0)}, M, \Delta \Big) =$$
$$\int p \Big( X_{\text{imp}}^{(m=1)} \, \Big| \, X_{\text{hist}}, X_{\text{imp}}^{(m=0)}, M, \Delta \Big)$$
$$\cdot \, p \Big( X_{\text{hist}} \, \Big| \, X_{\text{imp}}^{(m=0)}, M, \Delta \Big) \, dX_{\text{hist}}. \quad (2)$$

Excluding historical context amounts to marginalizing over $X_{\text{hist}}$, yielding a marginal conditional distribution that averages temporally distinct hypotheses and reduces temporal modeling to intra-window consistency.

To characterize this effect at the objective level, define the conditional expected risk over missing positions, indexed by the missing set $\Omega(M)$, with respect to an information set $\mathcal{I}$ as

$$\mathcal{R}(\theta \mid \mathcal{I}) = \mathbb{E}_{X_{\text{imp}}^{(m=1)} \sim p(\cdot \mid \mathcal{I})} \Big[ \ell_{\Omega(M)} \Big( X_{\text{imp}}^{(m=1)}, f_\theta(\mathcal{I}) \Big) \Big]. \quad (3)$$

Different imputation paradigms correspond to different choices of the information set $\mathcal{I}$, which in turn induce different conditional distributions $p(X_{\text{imp}}^{(m=1)} \mid \mathcal{I})$. Specifically,

$$\mathcal{I}_{\text{full}} = \{X_{\text{hist}}, X_{\text{imp}}^{(m=0)}, M, \Delta\}, \quad (4)$$

$$\mathcal{I}_{\text{loc}} = \{X_{\text{imp}}^{(m=0)}, M, \Delta\}. \quad (5)$$

When $\mathcal{I} = \mathcal{I}_{\text{full}}$, the conditional distribution explicitly incorporates temporal evolution constraints; when $\mathcal{I} = \mathcal{I}_{\text{loc}}$, historical information $X_{\text{hist}}$ is implicitly marginalized, yielding a marginal (mixture) conditional distribution. Under the same loss function $\ell_{\Omega(M)}(\cdot)$, the achievable conditional risk therefore satisfies

$$\inf_{f_\theta} \mathcal{R}(\theta \mid \mathcal{I}_{\text{full}}) \leq \inf_{f_\theta} \mathcal{R}(\theta \mid \mathcal{I}_{\text{loc}}), \quad (6)$$

where $\inf$ denotes the infimum over imputation functions $f_\theta$. In summary, marginalizing $X_{\text{hist}}$ reduces temporal dependency modeling to local consistency, whereas explicitly conditioning on $X_{\text{hist}}$ preserves temporal dependencies at both the probabilistic and expected-risk levels.

## 4. The CBiT Architecture

### 4.1. Model Overview

As shown in Figure 3, CBiT is a hierarchical inference architecture for multivariate time-series imputation. Given the raw time-series input $\mathbf{X}$ and the corresponding missingness mask $\mathbf{M}$, the model first performs latent writing to distill long-range historical information and to initialize representations for imputation targets. Specifically, a Temporal Latent Writer maps the full observation space into two complementary token sets,

$$(\mathbf{Lat}, \ \mathbf{Imp}) = \mathcal{T}_{\text{latent}}(\mathbf{X}, \mathbf{M}), \quad (7)$$

where $\mathbf{Lat}$ denotes a compact set of latent tokens that encode compressed historical context over extended time horizons, and $\mathbf{Imp}$ denotes the initial token representations associated with the imputation-target variables. The operator $\mathcal{T}_{\text{latent}}(\cdot)$ corresponds to the Temporal Latent Writer, which jointly performs history compression and target initialization, thereby providing an efficient interface between raw observations and subsequent cross-block inference.

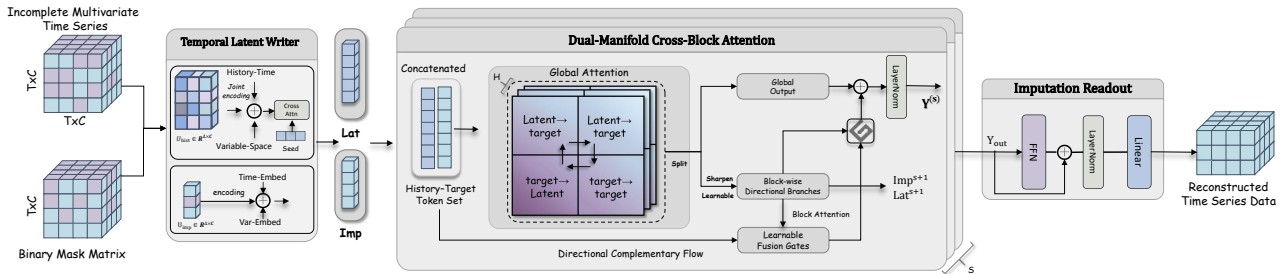

*Figure 3.* Model overview of the proposed CBiT, which consists of a Temporal Latent Writer for compressing long-range historical information into latent representations, and a Dual-Manifold Cross-Block Attention module for structured information interaction across blocks under distinct representation manifolds.

Based on these representations, CBiT performs cross-block inference across $S$ successive stages. At the $s$-th stage ($s = 1, \ldots, S$), historical latent tokens and observable representations of imputation-target variables are jointly updated via a Dual-Manifold Cross-Block Attention.

$$\left(\mathbf{Lat}^{(s+1)}, \mathbf{Imp}^{(s+1)}, \mathbf{Y}^{(s)}\right) = \mathcal{F}_{\mathrm{CBA}}^{(s)}\left(\mathbf{Lat}^{(s)}, \mathbf{Imp}^{(s)}\right), \quad (8)$$

where $\mathbf{Lat}^{(s+1)}$ denotes the updated historical latent tokens encoding compressed historical constraints, $\mathbf{Imp}^{(s+1)}$ denotes the refined token representations of imputation-target variables conditioned on historical context, and $\mathbf{Y}^{(s)}$ denotes the unified token representation that aggregates historical latent information and current variable states. Here, $\mathcal{F}_{\mathrm{CBA}}^{(s)}(\cdot)$ denotes the stage-wise *Dual-Manifold Cross-Block Attention* operator that jointly updates historical and imputation-target representations.

After the final inference stage, the model obtains a unified representation $\mathbf{Y}_{\mathrm{out}}$ that aligns global inference with the learned states of both the historical and imputation-target branches.

$$\hat{\mathbf{Y}} = \mathrm{Linear}\left(\mathrm{LN}\left(\mathbf{Y}_{\mathrm{out}} + \mathrm{FFN}\left(\mathbf{Y}_{\mathrm{out}}\right)\right)\right), \quad (9)$$

where $\mathrm{FFN}(\cdot)$ denotes a position-wise feed-forward network, $\mathrm{LN}(\cdot)$ denotes layer normalization, $\mathrm{Linear}(\cdot)$ denotes a linear projection, and $\hat{\mathbf{Y}}$ denotes the imputation outputs for the target variables.

### 4.2. Temporal Latent Writer

Modeling long historical context is critical for time-series imputation. However, directly applying self-attention over a history window of length $L$ with $C$ variables operates on $L \times C$ time–variable tokens and incurs prohibitive computational cost. To address this issue, CBiT first writes historical observations into a compact set of latent tokens with $O(NLC)$ cost, and then performs subsequent inference within the latent space with $O(N^2)$ complexity. This avoids applying full time–variable self-attention over all $L \cdot C$ tokens, whose complexity is $O((LC)^2)$, where $N \ll L \cdot C$.

Given a multivariate time series $\mathbf{X} = \{x_t\}_{t=1}^T \in \mathbb{R}^{T \times C}$ and its corresponding missingness mask $\mathbf{M} = \{m_t\}_{t=1}^T \in \{0, 1\}^{T \times C}$, where $T$ denotes the sequence length and $C$ denotes the number of variables, we first perform value–mask joint encoding over the full observation space, and then partition the resulting representations into historical and imputation-target components. Specifically, a shared encoder is applied to both $\mathbf{X}$ and $\mathbf{M}$ to obtain joint representations $\mathbf{U} = \{\mathbf{u}_t\}_{t=1}^T$, where

$$\mathbf{u}_t = \phi_x(x_t) + \phi_m(m_t), \quad (10)$$

with $\phi_x(\cdot)$ and $\phi_m(\cdot)$ denoting value and mask embedding functions, respectively. Based on the temporal partition induced by the inference stage, $\mathbf{U}$ is further decomposed into a historical component $\mathbf{U}_{\mathrm{hist}}$ and an imputation-target component $\mathbf{U}_{\mathrm{imp}}$. The history representations are then constructed as

$$\mathbf{E}^{\mathrm{hist}} = \mathbf{U}_{\mathrm{hist}} + \mathbf{e}_{\mathrm{time}} + \mathbf{e}_{\mathrm{var}} \in \mathbb{R}^{L \times C \times D}, \quad (11)$$

where $L$ denotes the length of the historical segment and $D$ denotes the embedding dimensionality, $\mathbf{e}_{\mathrm{time}} \in \mathbb{R}^{L \times C \times D}$ denotes history-aware temporal positional encodings that ensure consistent time semantics under adaptive-length history windows via relative normalization and multi-scale Fourier bases, and $\mathbf{e}_{\mathrm{var}} \in \mathbb{R}^{L \times C \times D}$ denotes variable-space encodings capturing inter-variable heterogeneity and window-dependent state differences through dynamic statistics and static variable identities. $\mathbf{Imp} \in \mathbb{R}^{r \times D}$ is constructed from $\mathbf{U}_{\mathrm{imp}}$ via a lightweight encoding scheme with linear time and space complexity, where $r$ denotes the number of observed tokens in the imputation-target segment, each corresponding to a time–variable pair.

The history representations are then flattened into $\mathbf{E}_{\mathrm{flat}} \in \mathbb{R}^{(L \cdot C) \times D}$ and written into a set of $N$ latent tokens via Perceiver-style cross-attention:

$$\mathbf{Lat} = \mathrm{CrossAttn}\left(Q = \mathbf{Lat}^{(0)}, \ K = \mathbf{E}_{\mathrm{flat}}, \ V = \mathbf{E}_{\mathrm{flat}}\right), \quad (12)$$

where $\mathbf{Lat}^{(0)} \in \mathbb{R}^{N \times D}$ denotes learnable latent queries. By compressing the history into $\mathbf{Lat}$, CBiT avoids full

self-attention over all historical time–variable tokens, while providing compact history-conditioned representations for subsequent cross-block inference.

## 4.3. Dual-Manifold Cross-Block Attention

Under CTIP, historical context and imputation targets play asymmetric roles. We introduce Dual-Manifold Cross-Block Attention, which computes global attention once and subsequently separates and supplements the inferred dependencies for historical context and imputation targets. This separation is performed as a post-attention resolution step, introducing only lightweight block-wise overhead.

### 4.3.1. DUAL-MANIFOLD TOKEN REPRESENTATION.

At inference stage $s$, the model maintains two semantically distinct token manifolds. The historical conditional manifold consists of $N$ latent tokens $\mathbf{Lat}^{(s)} = \{\mathbf{lat}_n^{(s)}\}_{n=1}^{N}$, where each token $\mathbf{lat}_n^{(s)} \in \mathbb{R}^D$ encodes compressed global contextual evidence from the historical window. In parallel, the model maintains a set of $r$ imputation-target tokens $\mathbf{Imp}^{(s)} = \{\mathbf{imp}_i^{(s)}\}_{i=1}^{r}$, with each $\mathbf{imp}_i^{(s)} \in \mathbb{R}^D$ representing an uncertain variable state to be inferred at the current stage.

At inference stage $s$, we construct a global relation space to capture dependencies between historical conditional tokens and imputation-target tokens.

$$\mathbf{Z}^{(s)} = \left[\mathbf{Lat}^{(s)}; \mathbf{Imp}^{(s)}\right] = \left\{\mathbf{z}_n^{(s)}\right\}_{n=1}^{N+r}. \quad (13)$$

We define $\mathcal{I}_{\text{hist}} = \{1, \ldots, N\}$ and $\mathcal{I}_{\text{cur}} = \{N+1, \ldots, N+r\}$ as the index sets of historical latent tokens and imputation-target tokens within the unified sequence, respectively.

### 4.3.2. GLOBAL ATTENTION RELATION MODELING.

Given the unified token sequence $\mathbf{Z}^{(s)} = \{\mathbf{z}_i^{(s)}\}_{i=1}^{N+r}$, each token is projected to query, key, and value representations under attention head $h$ as

$$\mathbf{q}_i^{(s,h)} = \mathbf{W}_Q^{(h)}\mathbf{z}_i^{(s)}, \ \mathbf{k}_i^{(s,h)} = \mathbf{W}_K^{(h)}\mathbf{z}_i^{(s)}, \ \mathbf{v}_i^{(s,h)} = \mathbf{W}_V^{(h)}\mathbf{z}_i^{(s)}, \quad (14)$$

where $\mathbf{q}_i^{(s,h)}$, $\mathbf{k}_i^{(s,h)}$, and $\mathbf{v}_i^{(s,h)}$ denote the query, key, and value vectors associated with the $i$-th token under the $h$-th attention head, respectively; $\mathbf{W}_Q^{(h)}$, $\mathbf{W}_K^{(h)}$, and $\mathbf{W}_V^{(h)} \in \mathbb{R}^{D \times d}$ are learnable projection matrices for the corresponding head; $D$ denotes the model dimensionality, $H$ is the total number of attention heads, and $d = D/H$ is the per-head feature dimension.

Based on these projections, the global attention weight is defined as

$$A_{i,j}^{(s,h)} = \frac{\exp\left(\langle \mathbf{q}_i^{(s,h)}, \mathbf{k}_j^{(s,h)}\rangle/\sqrt{d}\right)}{\sum\limits_{j' \in \mathcal{I}_{\text{hist}} \cup \mathcal{I}_{\text{cur}}} \exp\left(\langle \mathbf{q}_i^{(s,h)}, \mathbf{k}_{j'}^{(s,h)}\rangle/\sqrt{d}\right)}. \quad (15)$$

Here, indices $i$ and $j$ enumerate the unified token set $\mathcal{I}_{\text{hist}} \cup \mathcal{I}_{\text{cur}}$, corresponding to historical latent tokens and imputation-target tokens, respectively. The resulting attention matrix $\mathbf{A}^{(s,h)} = \left[A_{i,j}^{(s,h)}\right] \in \mathbb{R}^{(N+r)\times(N+r)}$ encodes the global relational structure shared by both manifolds.

### 4.3.3. CROSS-BLOCK SEMANTIC DECOMPOSITION.

We reuse the global attention matrix $\mathbf{A}^{(s,h)}$ and decompose attention relations $i \to j$, where token $i$ attends to token $j$, into direction-aware semantic blocks based on whether the query and attended tokens originate from the historical or imputation-target manifold, covering all intra- and cross-manifold interactions. At the matrix level, this directional factorization corresponds to a four-block partition of the global attention matrix:

$$\mathbf{A}^{(s,h)} = \begin{bmatrix} \mathbf{A}_{\text{hist}\to\text{hist}}^{(s,h)} & \mathbf{A}_{\text{hist}\to\text{cur}}^{(s,h)} \\ \mathbf{A}_{\text{cur}\to\text{hist}}^{(s,h)} & \mathbf{A}_{\text{cur}\to\text{cur}}^{(s,h)} \end{bmatrix}. \quad (16)$$

where hist and cur denote the historical latent tokens and the imputation-target tokens, respectively.

This decomposition only reorganizes attention relations. For each (in $\to$ out) block, value representations are derived from the out tokens via a direction-specific linear projection, where $\{\text{in}, \text{out}\} \in \{\text{hist}, \text{cur}\}$ indicate the two token manifolds.

$$\mathbf{v}_{\text{in}\to\text{out}}^{(s,h)} = \mathbf{W}_{\text{in}\to\text{out}}^{(h)} \bullet_{\text{out}}^{(s)}, \quad (17)$$

where $\bullet_{\text{out}}^{(s)} \in \{\mathbf{Lat}^{(s)}, \mathbf{Imp}^{(s)}\}$ depends on the semantic role of the attended tokens.

### 4.3.4. BLOCK-WISE ATTENTION WITH LEARNABLE PARAMETERIZATION

Although the global attention matrix is shared, cross-block relations differ in concentration, with historical relations being diffuse and imputation-target relations sparse. Uniform normalization across blocks can blur such block-specific structures. To address this, we apply block-wise adaptive calibration to each directional attention block. Formally, for each directional block $\mathbf{A}_{\text{in}\to\text{out}}^{(s,h)} \in \mathcal{B}$, the calibrated attention response is defined as

$$\widetilde{\mathbf{A}}_{\text{in}\to\text{out}}^{(s,h)} = \mathcal{C}_\alpha\left(\mathbf{A}_{\text{in}\to\text{out}}^{(s,h)}\right), \quad (18)$$

where $\mathcal{C}_\alpha(\cdot)$ denotes a block-specific calibration operator that rescales attention distributions and enhances dominant relations.

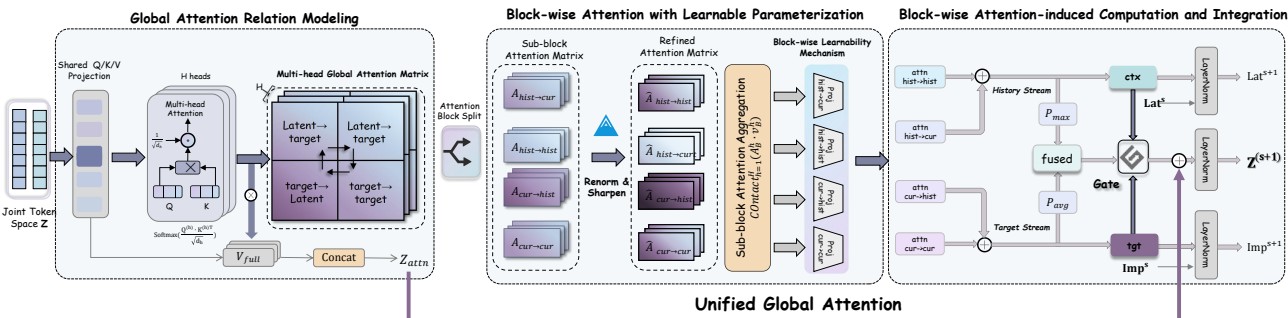

*Figure 4.* Overview of the Dual-Manifold Cross-Block Attention for cross-block information interaction.

For each directional relation block $\mathcal{B}$, the corresponding block-wise attention output at inference stage $s$ is defined as

$$\mathbf{attn}_{\mathcal{B}}^{(s)} = \mathbf{Proj}_{\mathcal{B}}^{(s)}\left(\text{Concat}_{h=1}^{H}\left(\sum_{(i,j)\in\mathcal{B}}\widetilde{A}_{i,j}^{(s,h)}\,\mathbf{v}_j^{(s,h)}\right)\right),$$

$$(19)$$

where $\mathbf{v}_j^{(s,h)}$ is the value representation associated with the attended token $j$ under attention head $h$, $\text{Concat}_{h=1}^{H}(\cdot)$ denotes concatenation along the head dimension, and $\mathbf{Proj}_{\mathcal{B}}^{(s)}$ represents a learnable linear projection specific to relation block $\mathcal{B}$ at stage $s$.

By introducing block-specific, learnable $\mathbf{Proj}_{\mathcal{B}}$ for different directional blocks, the design prevents block-level gradients from being confined to the shared global attention matrix. Each block is updated through its own projection parameters, enabling block-wise specialization under a unified global attention structure.

### 4.4. Lat–Imp Response Aggregation

Based on the direction-aware attention blocks, the model first constructs the principal response representations for the historical conditional manifold and the imputation-target manifold:

$$\mathbf{ctx}^{(s)} = \mathbf{attn}_{\text{hist}\to\text{hist}}^{(s)} + \mathbf{attn}_{\text{hist}\to\text{cur}}^{(s)},$$
$$\mathbf{tgt}^{(s)} = \mathbf{attn}_{\text{cur}\to\text{cur}}^{(s)} + \mathbf{attn}_{\text{cur}\to\text{hist}}^{(s)}.$$

$$(20)$$

where $\mathbf{ctx}^{(s)}$ and $\mathbf{tgt}^{(s)}$ denote stage-wise intermediate update variables produced by the historical and target branches, respectively.

The historical and target representations are then updated via residual connections followed by layer normalization:

$$\mathbf{Lat}^{(s+1)} = \text{LN}\left(\mathbf{Lat}^{(s)} + \mathbf{ctx}^{(s)}\right),$$
$$\mathbf{Imp}^{(s+1)} = \text{LN}\left(\mathbf{Imp}^{(s)} + \mathbf{tgt}^{(s)}\right).$$

$$(21)$$

To capture stage-level semantics, history-related responses and target-related responses are aggregated using different pooling strategies:

$$\mathbf{ctx}_{\text{agg}}^{(s)} = \mathcal{P}_{\max}\left(\mathbf{attn}_{\text{hist}\to\text{hist}}^{(s)}, \mathbf{attn}_{\text{hist}\to\text{cur}}^{(s)}\right),$$
$$\mathbf{tgt}_{\text{agg}}^{(s)} = \mathcal{P}_{\text{avg}}\left(\mathbf{attn}_{\text{cur}\to\text{cur}}^{(s)}, \mathbf{attn}_{\text{cur}\to\text{hist}}^{(s)}\right).$$

$$(22)$$

where $\mathcal{P}_{\max}(\cdot)$ and $\mathcal{P}_{\text{avg}}(\cdot)$ denote max- and average-based aggregation operators, respectively, and $\mathbf{ctx}_{\text{agg}}^{(s)}$ and $\mathbf{tgt}_{\text{agg}}^{(s)}$ are the corresponding aggregated intermediate variables from the historical and target branches.

The two aggregated summaries are first combined to form a shared global semantic representation:

$$\mathbf{fused}^{(s)} = \mathbf{ctx}_{\text{agg}}^{(s)} + \mathbf{tgt}_{\text{agg}}^{(s)}.$$

$$(23)$$

The principal responses are then integrated with the global semantic representation through a token-wise gating mechanism. The gating weights are defined as

$$\boldsymbol{\zeta}^{(s)} = \sigma\left(\mathbf{G}\left([\mathbf{ctx}^{(s)};\mathbf{tgt}^{(s)}]\right)\right),$$

$$(24)$$

where $\mathbf{G}(\cdot)$ denotes a learnable projection and $\sigma(\cdot)$ is the sigmoid function.

The gated fusion result is given by

$$\mathbf{fused}_{\text{gate}}^{(s)} = \boldsymbol{\zeta}^{(s)}\odot\left[\mathbf{ctx}^{(s)};\mathbf{tgt}^{(s)}\right] + \left(1-\boldsymbol{\zeta}^{(s)}\right)\odot\mathbf{fused}^{(s)},$$

$$(25)$$

where $\mathbf{fused}_{\text{gate}}^{(s)}$ denotes a stage-wise intermediate fused state produced by gated combination.

Finally, the unified representation for the next inference stage is obtained by

$$\mathbf{Y}^{(s)} = \text{LN}(\mathbf{fused}_{\text{gate}}^{(s)} + \mathbf{Z}_{\text{attn}}^{(s)})$$

$$(26)$$

where $\mathbf{Z}_{\text{attn}}^{(s)} = \text{Concat}_{h=1}^{H}\left(\mathbf{A}^{(s,h)}\mathbf{v}^{(s,h)}\right)$ denotes the unified token-level relational response of global attention.

## 5. Experiments

This section systematically evaluates the proposed model on multivariate time-series imputation tasks. Beyond reporting

*Table 1.* Effect of Joint Supervision on time-series imputation performance.

| | | Time series model | | | | Reconstruction Model | | | | Diffusion Model | | Unified Model | | Proposed | |
|---|---|---|---|---|---|---|---|---|---|---|---|---|---|---|---|
| | | DeformableTST | | XPatch | | SAITS | | ImputeFormer | | SSSD | | UNiTS | | CBiT | |
| | | MAE | RMSE | MAE | RMSE | MAE | RMSE | MAE | RMSE | MAE | RMSE | MAE | RMSE | MAE | RMSE |
| ETTh1 | 10% | 0.3829 | 0.5492 | 0.4329 | 0.6118 | 0.2467 | 0.3686 | 0.3715 | 0.5753 | 0.2369 | 0.3331 | 0.3026 | 0.4574 | 0.1658 | 0.2849 |
| | 20% | 0.4071 | 0.5775 | 0.4704 | 0.6475 | 0.2730 | 0.4534 | 0.4058 | 0.6307 | 0.2374 | 0.3395 | 0.3900 | 0.6039 | 0.1750 | 0.2956 |
| | 30% | 0.4576 | 0.6371 | 0.5048 | 0.6917 | 0.3110 | 0.5138 | 0.4070 | 0.6469 | 0.2986 | 0.4223 | 0.4733 | 0.7029 | 0.1959 | 0.3282 |
| | 40% | 0.4789 | 0.6507 | 0.5504 | 0.7477 | 0.4024 | 0.6401 | 0.7784 | 1.0210 | 0.3803 | 0.5165 | 0.5707 | 0.8362 | 0.2068 | 0.3411 |
| | 50% | 0.4299 | 0.6067 | 0.5996 | 0.8078 | 0.3729 | 0.6096 | 0.7756 | 1.0167 | 0.4144 | 0.5709 | 0.6304 | 0.9076 | 0.2217 | 0.3670 |
| ETTh2 | 10% | 0.1879 | 0.2661 | 0.2123 | 0.2911 | 0.1993 | 0.2661 | 0.1883 | 0.2635 | 0.2877 | 0.3765 | 0.1479 | 0.2175 | 0.0999 | 0.1477 |
| | 20% | 0.1846 | 0.2637 | 0.2303 | 0.3130 | 0.1540 | 0.2048 | 0.1639 | 0.2266 | 0.3165 | 0.4143 | 0.1970 | 0.2787 | 0.1038 | 0.1524 |
| | 30% | 0.2082 | 0.2889 | 0.2681 | 0.3669 | 0.2038 | 0.2662 | 0.1696 | 0.2330 | 0.4154 | 0.5223 | 0.2192 | 0.3079 | 0.1198 | 0.1798 |
| | 40% | 0.2390 | 0.3269 | 0.3146 | 0.4310 | 0.2220 | 0.2837 | 0.1932 | 0.2706 | 0.4164 | 0.5692 | 0.2496 | 0.3493 | 0.1255 | 0.1877 |
| | 50% | 0.2403 | 0.3261 | 0.3570 | 0.4846 | 0.2127 | 0.2824 | 0.2065 | 0.2853 | 0.5283 | 0.6839 | 0.2780 | 0.3849 | 0.1354 | 0.1985 |
| WTH | 10% | 0.2218 | 0.4342 | 0.2568 | 0.4482 | 0.1242 | 0.3652 | 0.1390 | 0.3893 | 0.2275 | 0.4268 | 0.2309 | 0.4527 | 0.1165 | 0.3285 |
| | 20% | 0.2442 | 0.4616 | 0.2836 | 0.4682 | 0.1218 | 0.3595 | 0.1337 | 0.3975 | 0.1775 | 0.3899 | 0.2158 | 0.4378 | 0.1196 | 0.3381 |
| | 30% | 0.2442 | 0.4519 | 0.3266 | 0.5118 | 0.1362 | 0.3694 | 0.1552 | 0.3880 | 0.2995 | 0.4910 | 0.2482 | 0.4708 | 0.1273 | 0.3423 |
| | 40% | 0.2583 | 0.4650 | 0.3972 | 0.5859 | 0.1627 | 0.4111 | 0.1578 | 0.4189 | 0.3198 | 0.5203 | 0.3267 | 0.5644 | 0.1463 | 0.3745 |
| | 50% | 0.2410 | 0.4638 | 0.4014 | 0.6047 | 0.1514 | 0.4292 | 0.1748 | 0.4255 | 0.2760 | 0.4964 | 0.4220 | 0.6756 | 0.1487 | 0.3863 |
| Power | 10% | 0.0542 | 0.1136 | 0.1015 | 0.1583 | 0.0841 | 0.1271 | 0.0485 | 0.1163 | 0.0590 | 0.0956 | 0.0693 | 0.1537 | 0.0336 | 0.0874 |
| | 20% | 0.0673 | 0.1205 | 0.1351 | 0.2023 | 0.1048 | 0.1763 | 0.0520 | 0.1174 | 0.0713 | 0.1126 | 0.1107 | 0.2291 | 0.0347 | 0.0901 |
| | 30% | 0.0775 | 0.1384 | 0.1820 | 0.2628 | 0.1370 | 0.2080 | 0.0682 | 0.1666 | 0.0785 | 0.1274 | 0.1809 | 0.3336 | 0.0397 | 0.1045 |
| | 40% | 0.2122 | 0.3934 | 0.2492 | 0.3523 | 0.1356 | 0.2334 | 0.0598 | 0.1413 | 0.1383 | 0.2120 | 0.3307 | 0.5313 | 0.0454 | 0.1137 |
| | 50% | 0.2573 | 0.3990 | 0.3027 | 0.4356 | 0.1862 | 0.2992 | 0.0676 | 0.1560 | 0.1182 | 0.1904 | 0.4264 | 0.6204 | 0.0557 | 0.1502 |

final imputation accuracy, we conduct comparative experiments under two training settings—joint supervision and missing-only supervision—across multiple datasets and representative imputation methods, in order to assess model effectiveness and stability when training signals are provided exclusively at missing positions. Additional experimental results and analyses are provided in the Appendix.

## 5.1. Experimental Setup

### 5.1.1. DATASETS

We conduct experiments on several public multivariate time-series datasets, including the ETT datasets (ETTh1, ETTh2, ETTm1, ETTm2) (Zhou et al., 2021), Air Quality(Air) (Vito, 2008), Power (Salam & El Hibaoui, 2018), Weather[1](WTH), and Solar (Alexandrov et al., 2020).

To systematically evaluate the imputation capability of different models under varying levels of missingness, the missing ratios are set from 10% to 50% across experiments.

### 5.1.2. BASELINES

We compare our method with representative approaches from different categories, including time-series models (XPatch (Stitsyuk & Choi, 2025), DeformableTST (Luo & Wang, 2024)), reconstruction-based imputation models (SAITS (Du et al., 2023), ImputeFormer (Nie et al., 2024)),

diffusion-based generative imputation models (SSSD (Alcaraz & Strodthoff, 2023)), and unified models supporting both forecasting and imputation (UNiTS (Gao et al., 2024)).

### 5.1.3. TRAINING OBJECTIVES

During training, we adopt mask-aware loss functions that explicitly distinguish supervision on missing and observed positions. Specifically, we consider two training settings:
(i) **Joint Supervision.** The model is supervised on both missing and observed positions. The training objective consists of the Masked MAE on missing entries and a weighted Observed MAE on observed entries, which serves as a reconstruction regularizer to improve training stability.
(ii) **Missing-only Supervision.** The model is trained solely with the Masked MAE on missing positions, without any reconstruction loss on observed entries. This setting evaluates whether the model can learn meaningful temporal and cross-variable structures using only imputation signals.

For validation and testing, imputation errors are consistently evaluated only on missing positions to ensure fair comparison across methods. In this setting, all results are evaluated using Masked MAE and Masked RMSE.

---

[1]Weather dataset was acquired at https://www.ncei.noaa.gov/data/local-climatological-data/

*Table 2.* Effect of Missing-only Supervision on time-series imputation performance.

| | | Time series model | | | | Reconstruction Model | | | | Diffusion Model | | Unified Model | | Proposed | |
|---|---|---|---|---|---|---|---|---|---|---|---|---|---|---|---|
| | | DeformableTST | | XPatch | | SAITS | | ImputeFormer | | SSSD | | UNiTS | | CBiT | |
| | | MAE | RMSE | MAE | RMSE | MAE | RMSE | MAE | RMSE | MAE | RMSE | MAE | RMSE | MAE | RMSE |
| ETTh1 | 10% | 0.4066 | 0.5853 | 0.4504 | 0.6379 | 0.3336 | 0.4672 | 0.4287 | 0.6913 | 0.2821 | 0.5767 | 0.6371 | 0.9591 | 0.2002 | 0.3432 |
| | 20% | 0.3903 | 0.5574 | 0.4690 | 0.6635 | 0.2568 | 0.4325 | 0.3735 | 0.5934 | 0.2362 | 0.4179 | 0.4683 | 0.6909 | 0.1862 | 0.3098 |
| | 30% | 0.3966 | 0.5595 | 0.4946 | 0.6871 | 0.3076 | 0.5219 | 0.7764 | 1.0180 | 0.2923 | 0.4317 | 0.5218 | 0.7588 | 0.2094 | 0.3400 |
| | 40% | 0.4140 | 0.5894 | 0.5151 | 0.7176 | 0.3288 | 0.5498 | 0.7783 | 1.0207 | 0.2724 | 0.4069 | 0.5793 | 0.8268 | 0.2145 | 0.3479 |
| | 50% | 0.4395 | 0.6311 | 0.5583 | 0.7762 | 0.3795 | 0.6210 | 0.7738 | 1.0140 | 0.3673 | 0.5306 | 0.6522 | 0.9112 | 0.2389 | 0.3756 |
| ETTh2 | 10% | 0.1989 | 0.2739 | 0.2128 | 0.2901 | 0.1869 | 0.2507 | 0.2011 | 0.2754 | 0.1781 | 0.4543 | 0.1680 | 0.2415 | 0.1597 | 0.2267 |
| | 20% | 0.1930 | 0.2700 | 0.2268 | 0.3092 | 0.1503 | 0.2054 | 0.4444 | 0.5587 | 0.2096 | 0.3754 | 0.1941 | 0.2747 | 0.1178 | 0.1691 |
| | 30% | 0.1873 | 0.2617 | 0.2354 | 0.3231 | 0.1779 | 0.2342 | 0.1711 | 0.2350 | 0.2086 | 0.3465 | 0.2421 | 0.3414 | 0.1261 | 0.1867 |
| | 40% | 0.1914 | 0.2664 | 0.2670 | 0.3672 | 0.1958 | 0.2703 | 0.5125 | 0.6070 | 0.1416 | 0.2021 | 0.2823 | 0.3975 | 0.1386 | 0.2010 |
| | 50% | 0.1891 | 0.2636 | 0.2987 | 0.4156 | 0.1890 | 0.2619 | 0.5326 | 0.6456 | 0.2507 | 0.3524 | 0.3554 | 0.4853 | 0.1648 | 0.2309 |
| WTH | 10% | 0.2279 | 0.4289 | 0.2605 | 0.4486 | 0.1377 | 0.3541 | 0.1461 | 0.3945 | 0.2507 | 0.6291 | 0.1820 | 0.4034 | 0.1164 | 0.3290 |
| | 20% | 0.2268 | 0.4342 | 0.2810 | 0.4678 | 0.1403 | 0.3639 | 0.1573 | 0.3851 | 0.2636 | 0.6066 | 0.2230 | 0.4374 | 0.1357 | 0.3620 |
| | 30% | 0.2284 | 0.4397 | 0.2931 | 0.4744 | 0.1393 | 0.3765 | 0.1478 | 0.4031 | 0.2264 | 0.4929 | 0.2700 | 0.4857 | 0.1241 | 0.3665 |
| | 40% | 0.2379 | 0.4481 | 0.3166 | 0.5035 | 0.1547 | 0.4038 | 0.1785 | 0.4249 | 0.2524 | 0.4893 | 0.3792 | 0.5903 | 0.1366 | 0.4279 |
| | 50% | 0.2455 | 0.4520 | 0.3316 | 0.5145 | 0.1815 | 0.4269 | 0.7734 | 0.9728 | 0.2624 | 0.5157 | 0.3896 | 0.6154 | 0.1777 | 0.4240 |
| Power | 10% | 0.0584 | 0.1088 | 0.1160 | 0.1727 | 0.0758 | 0.1163 | 0.0524 | 0.1215 | 0.1668 | 0.4956 | 0.0733 | 0.1418 | 0.0371 | 0.0891 |
| | 20% | 0.0589 | 0.1078 | 0.1499 | 0.2136 | 0.0878 | 0.1401 | 0.0471 | 0.1168 | 0.0565 | 0.0974 | 0.1108 | 0.2092 | 0.0368 | 0.0939 |
| | 30% | 0.0580 | 0.1089 | 0.1890 | 0.2634 | 0.0870 | 0.1407 | 0.0596 | 0.1434 | 0.0444 | 0.1080 | 0.2835 | 0.4327 | 0.0427 | 0.1054 |
| | 40% | 0.0672 | 0.1203 | 0.2179 | 0.3012 | 0.0919 | 0.1539 | 0.0503 | 0.1263 | 0.0785 | 0.1307 | 0.3733 | 0.5450 | 0.0493 | 0.1149 |
| | 50% | 0.0720 | 0.1313 | 0.2815 | 0.3851 | 0.1832 | 0.2820 | 0.0538 | 0.1350 | 0.1014 | 0.1760 | 0.4525 | 0.6945 | 0.0596 | 0.1414 |

## 5.2. Experimental Results on Imputation under Joint Supervision.

Table 1 reports the imputation performance under joint supervision. Compared with existing imputation baselines, CBiT achieves consistently superior results across all datasets and missing ratios under both MAE and RMSE metrics. Specifically, on the ETTh1 dataset, CBiT reduces the MAE and RMSE by **35.4%** and **23.9%**, respectively, while on the Power dataset, the corresponding error reductions reach **29.49%** (MAE) and **13.96%** (RMSE).

*Table 3.* Computational efficiency comparison of imputation models.

| | FLOPS(G) | Params(M) |
|---|---|---|
| SAITS | 0.0774(G) | 0.2045(M) |
| ImputeFormer | 0.6090(G) | 0.5709(M) |
| UniTS | 2.2620(G) | 4.0968(M) |
| SSSD | 3.7352(G) | 24.443(M) |
| CBiT | 0.0586(G) | 0.2753(M) |

## 5.3. Experimental Results on Imputation under Missing-only Supervision.

Table 2 summarizes the results under missing-only supervision, where training signals are provided exclusively at missing positions, in contrast to the joint supervision setting reported above. Under this more challenging setting, CBiT demonstrates stronger performance than existing baseline methods under both MAE and RMSE metrics. Specifically, on the ETTh1 dataset, CBiT reduces the MAE and RMSE by **22.0%** and **23.5%**, respectively, while on the ETTh2 dataset, the corresponding reductions reach **13.6%** (MAE) and **11.3%** (RMSE).

## 5.4. Computational Efficiency and Model Complexity.

Table 3 summarizes the parameter counts and FLOPs of different imputation methods. CBiT uses **0.2753M** parameters and requires **0.0586G** FLOPs, achieving the second-lowest parameter count and the best computational efficiency among the compared methods. Considering imputation accuracy and efficiency jointly, CBiT provides the most favorable performance–efficiency trade-off.

## 6. Conclusion

In this work, we address a fundamental limitation of existing time-series imputation methods that rely on local conditional reconstruction and struggle to model long-range temporal dependencies. We introduce CTIP and develop CBiT as an effective instantiation for history-aware imputation under a unified temporal inference framework. Overall, our results show that conditional temporal inference yields more accurate and stable imputation than conventional reconstruction-based approaches.

## Acknowledgements

This work was supported by the National Key R&D Program of China (2024YFB3214500).

## Impact Statement

This work aims to improve the reliability of time-series imputation for incomplete multivariate data. The proposed method may benefit applications involving sensor monitoring, energy systems, and environmental analysis by improving data quality under missing observations. Potential risks include over-reliance on imputed values in high-stakes decision-making systems, where uncertainty should be carefully quantified and domain-specific validation remains necessary.

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

# A. Related Work

## A.1. Missing Data Imputation

Missing data imputation has long been a fundamental problem in time-series analysis. Existing approaches can be broadly categorized into three groups: traditional imputation methods, generative imputation models, and end-to-end deep imputation models.

**Traditional methods.** Early studies primarily relied on statistical and classical machine learning techniques (Goldani, 2024; Fuad Ahmad et al., 2024), such as KNN (Cover & Hart, 1967), SVR (Drucker et al., 1996), ARIMA (Box et al., 2015), and random forests (Breiman, 2001). These methods typically assume local similarity or stationarity and can be effective under low missing rates and simple dynamics. However, their performance degrades significantly in scenarios with high missing rates, strong nonlinearity, and complex temporal dependencies. Subsequent extensions, such as KNNI (Van Hulse & Khoshgoftaar, 2014) and BayoTIDE (Fang et al., 2024), aimed to improve robustness under incomplete observations, but still struggle to model complex temporal structures and scale to high-dimensional multivariate settings.

**Generative imputation models.** With the advancement of deep generative modeling, GAN- and diffusion-based imputation methods have been proposed (Zhang et al., 2025a; Dai et al., 2021; Oh et al., 2021), including GAIN (Yoon et al., 2018), CSDI (Tashiro et al., 2021), and MTSCI (Zhou et al., 2024). These approaches model conditional distributions of missing values and alleviate explicit distributional assumptions. Nevertheless, they often suffer from training instability, high inference cost, and limited ability to explicitly encode temporal structures.

**End-to-end time-series imputation models.** End-to-end imputation models have attracted sustained interest in time-series settings (Shan et al., 2023; Liu et al., 2019; Marisca et al., 2022; Cini et al., 2021), with representative methods such as BRITS (Cao et al., 2018), GRU-D (Che et al., 2018), and CDSA (Ma et al., 2019). These models directly operate on incomplete sequences using recurrent or attention-based mechanisms, yielding substantial performance improvements. However, most of them rely on local temporal windows or directional recurrence, which limits their ability to capture globally consistent temporal dynamics. Recent works incorporate forecasting-oriented designs to assist imputation (Challu et al., 2022; Tran et al., 2023; Kim et al., 2021), including TimeFlow (Naour et al., 2024) and TSRM (Leppich et al., 2025), but typically treat forecasting and imputation as loosely coupled tasks rather than unifying temporal dynamics within the imputation process itself.

## A.2. Time-Series Modeling

Time-series modeling remains a central topic in deep learning. Early approaches were largely based on recurrent neural networks, such as LSTM (Hochreiter & Schmidhuber, 1997), for short- and mid-term dependency modeling. More recently, Transformer-based models (Vaswani et al., 2017; Qiu et al., 2025) have achieved notable success due to their global attention mechanisms, while state space models (SSMs) and their efficient implementations, such as Mamba (Gu & Dao, 2024), further advanced long-sequence modeling.

Most existing time-series models focus on forecasting (Torres et al., 2021) and classification (Faouzi, 2022). A variety of Transformer- and SSM-based architectures have been proposed (Liu et al., 2023; Lin et al., 2021; Woo et al., 2022; Wang et al., 2024b; Chen et al., 2023; Challu et al., 2023), including FEDformer (Zhou et al., 2022) and Autoformer (Wu et al., 2021), which excel at capturing long-range dependencies and periodic patterns. For time-series classification, temporal dynamics have been extensively explored in diverse domains such as network security (Zhang et al., 2025b), leading to effective models (Yang et al., 2022; Cheng et al., 2023; Yeh et al., 2023; Chen et al., 2024). However, most of these models assume fully observed inputs and are not designed for missing-data scenarios, making their direct application to imputation structurally mismatched.

## A.3. Perceiver Models

In the computer vision field, attention-based architectures have been widely adopted across diverse visual tasks (Yang et al., 2025b; 2026; Zhong et al., 2024). To mitigate the quadratic complexity of self-attention on long sequences and high-dimensional inputs, Jaegle et al. proposed the Perceiver architecture (Jaegle et al., 2021b), which compresses inputs into a fixed number of latent queries. Originally developed as a general perception architecture, Perceiver demonstrated the effectiveness of latent bottlenecks for scalable representation learning over high-dimensional inputs. Subsequent variants,

including Perceiver-IO (Jaegle et al., 2021a) and Perceiver-VL (Tang et al., 2023), extended the framework to more complex multimodal settings (Carreira et al., 2022; Zhu et al., 2022). More recently, Perceiver-based models have been introduced to time-series domains, such as TimePerceiver (Lee & Lee, 2025) and related variants (Le et al., 2023), primarily to improve computational efficiency.

However, existing Perceiver-based time-series methods mainly focus on efficiency or predictive accuracy. Their latent compression mechanisms are not specifically designed to address the core challenges of time-series imputation, namely conditional inference and recovery under missing observations, and thus remain limited in modeling how historical context constrains current missing states.

## B. Experimental Setup

### B.1. Datasets and Settings

We evaluate the proposed model on multiple public multivariate time-series datasets covering diverse application domains, including energy systems, environmental monitoring and meteorology. These datasets exhibit heterogeneous data distributions, temporal resolutions, and variable dependencies, allowing a comprehensive assessment of imputation performance. The datasets used in our experiments are summarized as follows:

- **ETT (Electricity Transformer Temperature)** (Zhou et al., 2021): The ETT dataset is a widely used benchmark for long-term planning and operation in electric power systems. It contains two years of measurements collected from two counties in China. We adopt {ETTh1, ETTh2} with an hourly sampling rate and {ETTm1, ETTm2} with a 15-minute sampling rate. Each time step consists of one target variable (transformer oil temperature) and six power load–related covariates.

- **Air Quality**(Air) (Vito, 2008): An environmental monitoring dataset collected from a multisensor gas device deployed in an Italian city. Hourly averaged sensor responses are recorded together with reference gas concentrations measured by certified analyzers.

- **Power** (Salam & El Hibaoui, 2018): A regional electricity consumption dataset from Tetouan, Morocco, where multiple zones exhibit strong cross-variable correlations as well as pronounced daily and weekly periodic patterns.

- **Weather**[1](WTH): This dataset contains local climatological observations from nearly 1,600 locations in the United States, spanning four years from 2010 to 2013. Data are collected at an hourly frequency, with each time step including the target variable *wet bulb temperature* and 11 meteorological features.

- **Solar** (Alexandrov et al., 2020): A multivariate solar power generation benchmark provided by GluonTS. The dataset exhibits strong diurnal and seasonal variations and is commonly used to evaluate the generalization of time-series models in energy-related scenarios.

All datasets are normalized before training. The data are split into training, validation, and test sets in chronological order. Since CTIP requires access to preceding historical context at inference time, we provide the same pre-test history to all methods for consistency and fair comparison.

### B.2. Baselines

To ensure a fair and comprehensive comparison, we evaluate our method against representative baselines from different imputation categories, including statistical methods(MICE, MissForest), time-series models(XPatch, DeformableTST), reconstruction-based approaches(SAITS, ImputeFormer), and generative imputation models(SSSD),unified models supporting both forecasting and imputation(UNiTS):

- **MICE** (Van Buuren & Groothuis-Oudshoorn, 2011): A multiple imputation method based on chained equations, which models inter-variable statistical dependencies but has limited capability in capturing temporal structures.

- **MissForest** (Stekhoven & Bühlmann, 2012): A nonparametric imputation method based on random forests, capable of modeling nonlinear relationships but without explicit temporal modeling.

- **XPatch** (Stitsyuk & Choi, 2025): A patch-based Transformer model for time-series analysis, which captures long-range temporal dependencies and is adapted for imputation via prediction of missing entries.

- **DeformableTST** (Luo & Wang, 2024): A Transformer variant with deformable attention, allowing adaptive selection of informative historical positions for complex temporal modeling.

- **SAITS** (Du et al., 2023): A reconstruction-based model specifically designed for multivariate time-series imputation, which iteratively recovers missing values using stacked self-attention layers.

- **ImputeFormer** (Nie et al., 2024): A Transformer-based imputation model incorporating low-rank inductive biases to improve generalization in high-dimensional settings.

- **SSSD** (Alcaraz & Strodthoff, 2023): A generative imputation approach that combines structured state-space models with diffusion processes to model continuous-time dynamics.

- **UNiTS** (Gao et al., 2024): A unified multitask time-series model capable of handling forecasting, imputation, and other tasks within a single architecture.

## B.3. Evaluation Metrics and Training Objectives

**Evaluation Metrics**   Let $\mathbf{X} \in \mathbb{R}^{T \times C}$ denote the complete time series, $\hat{\mathbf{X}}$ the imputed output produced by the model, and $\mathbf{M} \in \{0, 1\}^{T \times C}$ the missingness mask, where $M_{t,c} = 1$ indicates a missing entry and $M_{t,c} = 0$ denotes an observed one. We adopt the mean absolute error (MAE) and root mean squared error (RMSE) computed exclusively over missing positions as evaluation metrics for imputation performance, defined as:

$$\text{Masked-MAE} = \frac{\sum_{t,c} M_{t,c} \, |\hat{X}_{t,c} - X_{t,c}|}{\sum_{t,c} M_{t,c}}, \tag{27}$$

$$\text{Masked-RMSE} = \sqrt{\frac{\sum_{t,c} M_{t,c} \, (\hat{X}_{t,c} - X_{t,c})^2}{\sum_{t,c} M_{t,c}}}. \tag{28}$$

These metrics are evaluated only on missing entries, measuring the accuracy of recovered values.

**Training Objectives**   During training, we consider two supervision settings under different experimental configurations. We first define the imputation loss on missing positions (Masked MAE):

$$\mathcal{L}_{\text{masked}} = \frac{\sum_{t,c} M_{t,c} \, |\hat{X}_{t,c} - X_{t,c}|}{\sum_{t,c} M_{t,c}}, \tag{29}$$

and the reconstruction loss on observed positions (Observed MAE):

$$\mathcal{L}_{\text{obs}} = \frac{\sum_{t,c}(1 - M_{t,c}) \, |\hat{X}_{t,c} - X_{t,c}|}{\sum_{t,c}(1 - M_{t,c})}. \tag{30}$$

**(i) Missing-only Supervision.**  Under this setting, the model is optimized solely based on imputation errors at missing positions. The training objective is defined as:

$$\mathcal{L}_{\text{miss}} = \mathcal{L}_{\text{masked}}. \tag{31}$$

**(ii) Joint Supervision.** In this setting, the model optimizes imputation accuracy at missing positions while incorporating reconstruction errors at observed positions as an auxiliary regularization term. The joint training objective is defined as:

$$\mathcal{L}_{\text{joint}} = \mathcal{L}_{\text{masked}} + \lambda \mathcal{L}_{\text{obs}}, \tag{32}$$

where $\lambda$ is a weighting coefficient balancing imputation accuracy and reconstruction consistency.

Notably, under both training strategies, evaluation during validation and testing is performed exclusively on missing positions, with reconstruction errors on observed positions excluded from model comparison to ensure consistent and fair evaluation across different training settings.

**Baseline Training Strategy** To ensure fair and consistent comparison, all baseline models are trained using a unified loss formulation. For baseline methods that include imputation-specific auxiliary objectives, such as the explicit low-rank regularization in ImputeFormer, these original components are retained and jointly optimized with the unified imputation loss within the same training framework.

## B.4. Missingness Mechanisms

To comprehensively evaluate model robustness under different missing-data assumptions, we consider two widely adopted missingness mechanisms: *Missing Completely at Random (MCAR)* and *Missing at Random (MAR)*. These two settings correspond to fundamentally different conditional dependency structures between missingness patterns and the underlying data.

Let $\mathbf{X} \in \mathbb{R}^{T \times C}$ denote the complete time series and $\mathbf{M} \in \{0,1\}^{T \times C}$ the missingness mask, where $M_{t,c} = 1$ indicates that $X_{t,c}$ is missing.

### B.4.1. MISSING COMPLETELY AT RANDOM (MCAR).

Under the MCAR assumption, the missingness pattern is independent of both observed and unobserved data values. Formally, the missingness distribution satisfies

$$p(\mathbf{M} \mid \mathbf{X}) = p(\mathbf{M}), \tag{33}$$

which implies that every entry has an equal probability of being missing, regardless of its temporal context or variable value.

In practice, MCAR masks are generated by independently sampling each entry:

$$M_{t,c} \sim \text{Bernoulli}(\rho), \tag{34}$$

where $\text{Bernoulli}(\rho)$ denotes a Bernoulli distribution with success probability $\rho \in (0,1)$, corresponding to the predefined missing rate. This setting primarily evaluates a model's ability to exploit temporal and cross-variable dependencies when missingness itself carries no informative structure.

### B.4.2. MISSING AT RANDOM (MAR).

Under the MAR mechanism, the missingness of a variable may depend on other *observed* variables but not on its own unobserved value. Formally, for each target variable $c$, the missing indicator $M_{t,c}$ satisfies

$$\mathbb{P}(M_{t,c} = 1 \mid \mathbf{X}) = \mathbb{P}(M_{t,c} = 1 \mid \mathbf{X}_{t,\mathcal{C}_{\text{cond}}^{(c)}}), \tag{35}$$

where $M_{t,c} \in \{0,1\}$ denotes the missing indicator of variable $c$ at time $t$, and $\mathcal{C}_{\text{cond}}^{(c)} \subset \{1,\ldots,C\} \setminus \{c\}$ denotes a predefined set of conditioning variable indices for variable $c$, which are used to model MAR dependencies.

Concretely, for each target variable $c$ subject to MAR, we construct a conditional signal

$$s_t^{(c)} = \frac{1}{|\mathcal{C}_{\text{cond}}^{(c)}|} \sum_{c' \in \mathcal{C}_{\text{cond}}^{(c)}} X_{t,c'}, \tag{36}$$

where $s_t^{(c)}$ denotes the conditional signal for variable $c$ at time $t$, $|\mathcal{C}_{\text{cond}}^{(c)}|$ is the cardinality of the conditioning set, and $X_{t,c'}$ denotes the observed value of variable $c'$ at time $t$.

The conditional signal is then mapped to a missingness probability via a logistic function:

$$\mathbb{P}(M_{t,c} = 1) = \sigma\left(\alpha \cdot \frac{s_t^{(c)} - \mu_c}{\sigma_c} + \beta\right), \tag{37}$$

where $\sigma(\cdot)$ denotes the sigmoid function, $\mu_c$ and $\sigma_c$ are the mean and standard deviation of $s_t^{(c)}$, respectively, $\alpha > 0$ controls the sensitivity of the missingness probability to the conditional signal, and $\beta$ is an intercept term determined by the predefined missing rate $\rho \in (0,1)$ (e.g., $\beta = \log \frac{\rho}{1-\rho}$).

This MAR setting induces structured and data-dependent missing patterns, resulting in informative and non-uniform missingness that requires models to perform conditional inference for effective imputation.

## B.5. Hyperparameter Settings

The hyperparameter configurations used during training are summarized in Table 4, including default values and corresponding descriptions.

*Table 4.* Hyperparameter settings used in our experiments.

| Hyperparameter | Default Value | Description |
| --- | --- | --- |
| Dataset | ETTh1 | Dataset used for experiments |
| Model | CBiT | Backbone imputation model |
| Batch size | 4 | Number of samples per batch |
| Learning rate | $1 \times 10^{-3}$ | Initial learning rate |
| Training epochs | 200 | Maximum number of epochs |
| Early stopping patience | 30 | Patience for early stopping |
| Random seed | 42 | Random seed for reproducibility |
| Missing rate | 0.1 | Ratio of injected missing values |
| Missing mode | MCAR / MAR | Missingness mechanism |
| Loss type | MAE | Loss formulation used for training |
| Observed loss weight $\lambda$ | 0.5 | Weight of observed-position loss |
| Input length ($L$) | 48 | Length of input sequence |
| Imputation length ($L_{\mathrm{imp}}$) | 4 | Length of imputed segment |
| Minimum history length | 1 | Minimum available history |
| Maximum sequence length | 2048 | Maximum positional encoding length |
| Encoder layers | 1 | Number of encoder layers |
| Hidden dimension ($D$) | 128 | Model hidden size |
| Attention heads | 8 | Number of attention heads |
| FFN dimension | 256 | Hidden size of FFN |
| Dropout rate | 0.1 | Dropout probability |
| Activation function | GELU | Activation in FFN |
| Latent construction type | Variable-based | Latent construction strategy |
| Number of latent tokens ($M$) | 16 | Number of latent tokens |
| Latent cross-attention heads | 4 | Heads in latent cross-attention |
| Attention stages | 1 | Number of CBiT stages |
| Sharpening type | Polynomial | Attention sharpening method |
| Polynomial init ratio | 0.2 | Initial sharpening ratio |
| Top-$k$ ratio | 0.5 | Ratio used in top-$k$ sharpening |

# C. Module Design and Mechanism Analysis

## C.1. Temporal Latent Writer

Figure 5 provides an overview of the proposed Temporal Latent Writer architecture. The module encodes historical observations into structured latent representations via joint time–variable compression, which are subsequently used as conditional context for imputation through cross-attention. Imputation targets are encoded separately with temporal and variable representations.

### C.1.1. History-Conditioned Representation Construction

To address the limited temporal modeling capability of traditional interpolation methods, we formulate missing-value imputation as a *history-conditioned temporal inference problem*, where historical observations provide stable and informative temporal context for recovering missing states. Given a multivariate time window of length $T$, the observed sequence is denoted as $\mathbf{X} = \{x_t\}_{t=1}^{T}$, with the corresponding missingness pattern $\mathbf{M} = \{m_t\}_{t=1}^{T}$.

At the input representation level, we adopt a unified value–mask joint encoding scheme to jointly model observed values

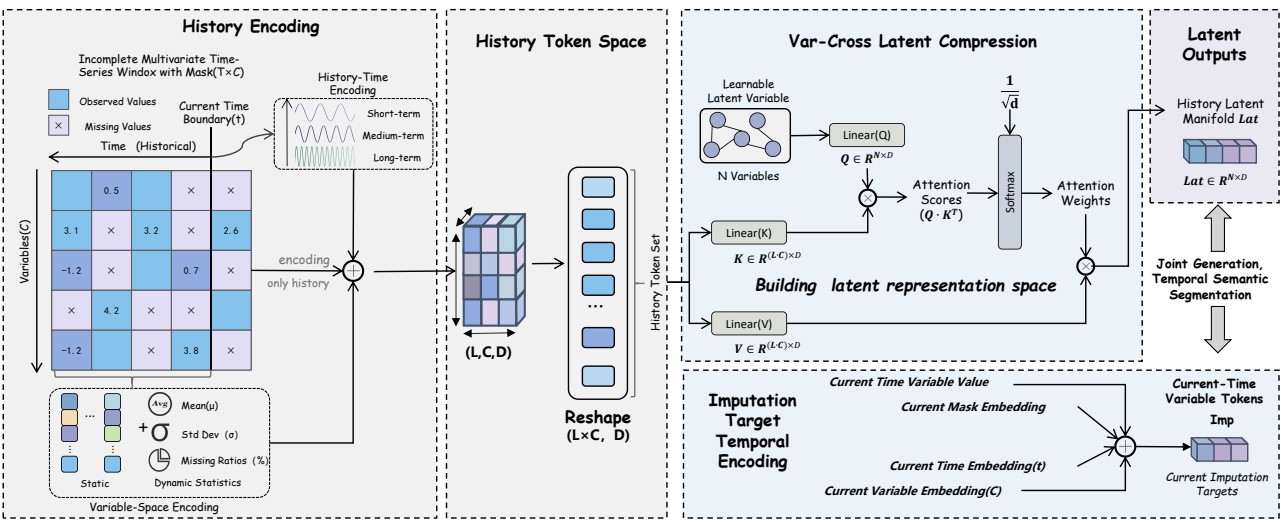

*Figure 5.* Temporal Latent Writer architecture.

and missing states. Specifically, for each time step $t$, the variable-level input representation is defined as

$$\mathbf{u}_t = \phi_x(x_t) + \phi_m(m_t), \qquad \mathbf{u}_t \in \mathbb{R}^{C \times D}, \tag{38}$$

where $\phi_x(\cdot)$ and $\phi_m(\cdot)$ denote the value encoding function and the missing-mask encoding function, respectively, $C$ is the number of variables, and $D$ is the unified latent feature dimension. For missing positions, the raw values are initialized to zero, and missingness information is explicitly conveyed to the model only through the mask encoding. This joint encoding mechanism maps *"what is observed"* and *"where it is missing"* into a shared representation space, transforming missingness from implicit uncertainty into an explicit, learnable structural signal.

Based on the unified encoding, the input sequence is partitioned into a historical segment and an imputation target segment. Let $L$ denote the length of the historical window and $\Delta$ the length of the imputation window. We then have

$$\mathbf{U}_{\text{hist}} = \{\mathbf{u}_t\}_{t=1}^{L}, \qquad \mathbf{U}_{\text{imp}} = \{\mathbf{u}_t\}_{t=L+1}^{L+\Delta}. \tag{39}$$

Although the two segments share the same representational form, they play distinct semantic roles: the former serves as conditional context, while the latter constitutes the inference target.

### C.1.2. VAR-CROSS LATENT COMPRESSION

Existing multivariate time-series models typically compress long sequences along the temporal dimension, mapping an $L \times C$ multivariate sequence into $N$ time-level tokens and performing modeling in the joint time–variable space with $\mathcal{O}((NC)^2)$ complexity. However, this strategy still incurs quadratic cost along the temporal dimension, which becomes computationally expensive in multivariate settings. Moreover, variable-wise structural dependencies are often prematurely aggregated and consequently weakened.

To further alleviate the computational burden of joint modeling, recent Perceiver-based approaches (Lee & Lee, 2025; Le et al., 2023) introduce a fixed number of latent queries to compress $L \times C$ time–variable tokens, thereby jointly accounting for temporal and variable dimensions. Nevertheless, in the context of time-series imputation, such methods tend to capture global statistical similarity or coarse temporal patterns, rather than explicitly modeling historical sequences as *conditional evidence* for inferring current missing states.

To address these limitations, we propose *Var-cross Latent Compression*, which constructs structured latent representations in the joint time–variable token space specifically for imputation-oriented inference. These latents are designed to capture cross-time and cross-variable dependencies and to directly constrain the missing-value recovery process. The compression is achieved via $N \ll L \cdot C$ latent queries, leading to a backbone computational complexity of $\mathcal{O}(N^2)$. Meanwhile, the introduced history-aware temporal encoding and variable-space encoding introduce only a lightweight, approximately linear overhead, which is negligible relative to the latent compression backbone. This design thus balances computational efficiency with structured imputation modeling.

**History-Time Encoding.** To construct consistent temporal semantics under variable-length historical windows, we normalize historical time indices into a relative coordinate system and generate time positional encodings using Fourier basis functions. Given a historical window of length $L$, for time index $t \in \{1, \dots, L\}$, we define the relative temporal coordinate as $\tau_t = \frac{t-L}{L}$, with $\tau_t \in [-1, 0]$. Based on this relative time axis, the historical time encoding is defined as

$$\mathbf{e}_{\text{time}}(t) = \mathbf{W}_t \Big[ \sin(2\pi k \tau_t), \ \cos(2\pi k \tau_t) \Big]_{k=1}^{K_{\text{eff}}} \in \mathbb{R}^D, \tag{40}$$

where $k$ denotes the frequency index, $K_{\text{eff}}$ is the effective number of frequencies adaptively determined by the historical length, and $\mathbf{W}_t$ is a linear projection matrix. This time encoding is applied exclusively to historical tokens and is used in subsequent latent compression and cross-block attention computation.

**Variable-Space Encoding.** To characterize variable heterogeneity and state differences across variables within the current historical window, we design a variable-space encoding scheme driven jointly by dynamic statistics and static variable identity.

Specifically, within the historical window, we compute per-variable statistical features, including the mean $\boldsymbol{\mu} \in \mathbb{R}^C$, standard deviation $\boldsymbol{\sigma} \in \mathbb{R}^C$, and missing ratio $\boldsymbol{\rho} \in \mathbb{R}^C$. These statistics are concatenated along the variable dimension and mapped through a shared nonlinear transformation to obtain dynamic variable embeddings: $\mathbf{e}_{\text{dyn}} = \text{MLP}([\boldsymbol{\mu}, \boldsymbol{\sigma}, \boldsymbol{\rho}]) \in \mathbb{R}^{C \times D}$. Meanwhile, to capture long-term semantic attributes of variables, we introduce a time-invariant static identity embedding for each variable: $\mathbf{e}_{\text{stat}} = \text{Embed}(\{1, \dots, C\}) \in \mathbb{R}^{C \times D}$. The final variable-space encoding is obtained via element-wise summation:

$$\mathbf{e}_{\text{var}} = \mathbf{e}_{\text{dyn}} + \mathbf{e}_{\text{stat}} \in \mathbb{R}^{C \times D}. \tag{41}$$

The variable-level representations within the historical window are obtained by integrating the value–mask joint encoding, time encoding, and variable-space encoding:

$$\mathbf{E}_{\text{var}}^{\text{hist}} = \mathbf{U}_{\text{hist}} + \mathbf{e}_{\text{time}} + \mathbf{e}_{\text{var}} \in \mathbb{R}^{L \times C \times D}. \tag{42}$$

### C.2. Token-Cross Attention

**Var-latent Writing.** We employ a cross-attention mechanism (Jaegle et al., 2021b) to compress the joint time–variable representations within the historical window into a fixed number of latent tokens, thereby obtaining a global historical context representation.

Given the variable-level historical encoding $\mathbf{E}_{\text{var}}^{\text{hist}} \in \mathbb{R}^{L \times C \times D}$, we treat the temporal and variable dimensions as a unified index space and flatten each time–variable pair into an independent token. This results in a historical token sequence consisting of $L \cdot C$ tokens, each with feature dimension $D$.

We then introduce a set of learnable latent seeds as global query vectors, with initial representations denoted as $\mathbf{Lat}^{(0)} \in \mathbb{R}^{N \times D}$, where $N \ll L \cdot C$. These latent seeds do not encode explicit temporal or variable priors, and solely serve as information compression queries.

Cross-attention is subsequently applied to write historical information into the latent space:

$$\mathbf{Lat} = \text{CrossAttn}\big(Q = \mathbf{Lat}^{(0)}, \ K = \mathbf{E}_{\text{flat}}, \ V = \mathbf{E}_{\text{flat}}\big) \in \mathbb{R}^{N \times D}, \tag{43}$$

where $\mathbf{E}_{\text{flat}} \in \mathbb{R}^{(L \cdot C) \times D}$ denotes the flattened historical token sequence. The resulting Var-latent representation jointly compresses temporal and variable information from the historical window and serves as a global contextual memory for subsequent imputation.

For observable time–variable pairs in the imputation target segment, we adopt the same value–mask joint encoding scheme as used for the historical window. The resulting representations are organized into a set of imputation-target tokens, defined row-wise as

$$\mathbf{Imp} = \mathbf{U}_{\text{imp}} + \mathbf{e}_t(t) + \mathbf{e}_v(c) \in \mathbb{R}^{r \times D}, \tag{44}$$

where each row corresponds to an observable time–variable index $(t, c)$, $r$ denotes the number of observed time–variable tokens within the imputation window, $\mathbf{e}_t(t)$ is a learnable linear temporal encoding associated with time index $t$, and $\mathbf{e}_v(c)$ is a learnable linear variable encoding associated with variable index $c$.

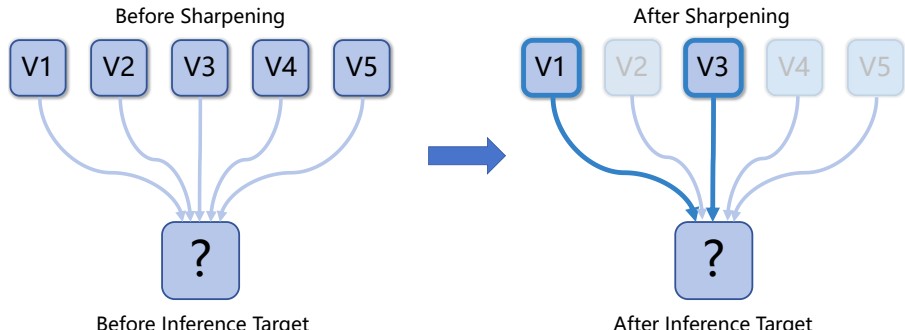

*Figure 6.* Block-wise attention sharpening. Attention distributions are re-calibrated within each directional block to emphasize important intra-block relations, yielding sharpened attention used for subsequent block-wise computation.

## C.3. Block-wise Attention Calibration

Let $\mathbf{A} \in \mathbb{R}^{N \times N}$ denote a cross-block attention matrix, where the $i$-th row represents the attention distribution of the $i$-th query token over all key tokens within the block. Before applying any sharpening operation, we first perform row-wise normalization on each attention block:

$$\tilde{A}_{ij} = \frac{A_{ij}}{\sum_{k=1}^{N} A_{ik} + \varepsilon}, \qquad \sum_{j=1}^{N} \tilde{A}_{ij} = 1. \tag{45}$$

On top of the normalized attention, we introduce two types of block-wise attention sharpening operators, which recalibrate the attention distributions within each directional block to concentrate mass on structurally important intra-block relations while suppressing noisy responses, thereby producing refined block-specific attention for subsequent relation aggregation, as illustrated in Figure 6.

### C.3.1. TOP-$k$ ATTENTION SHARPENING.

To explicitly suppress weak correlations within each attention block and retain only the most informative dependencies for imputation, we introduce a Top-$k$ attention sharpening strategy. Given a block-wise normalized attention matrix $\tilde{\mathbf{A}} \in \mathbb{R}^{N \times N}$, we restrict the support of each attention row to its $k$ largest entries.

The number of retained elements is determined by a power-law rule:

$$k = \lfloor N^{\rho} \rfloor + 1, \qquad \rho \in (0, 1], \tag{46}$$

where $N$ denotes the block size and $\rho$ controls the sparsity level of the sharpened attention.

Formally, let $\mathrm{Top}_k(\cdot)$ denote an operator that retains the $k$ largest elements of its input vector and sets all remaining entries to zero. The sharpened attention distribution is then obtained by restricting each row to its Top-$k$ entries followed by row-wise normalization:

$$\hat{\mathbf{A}}_{i,:} = \frac{\mathrm{Top}_k\left(\tilde{\mathbf{A}}_{i,:}\right)}{\sum_{j=1}^{N} \mathrm{Top}_k\left(\tilde{A}_{i,j}\right) + \varepsilon}, \tag{47}$$

where $\varepsilon$ is a small constant added for numerical stability.

### C.3.2. POLYNOMIAL ATTENTION SHARPENING.

As a continuous and learnable alternative, we introduce a second-order polynomial enhancement to the normalized attention weights, which amplifies high-confidence responses while relatively suppressing low-weight entries. Specifically, we define

$$\bar{A}_{ij} = \tilde{A}_{ij}\left(1 + \alpha \, \tilde{A}_{ij}\right) = \tilde{A}_{ij} + \alpha \, \tilde{A}_{ij}^2, \tag{48}$$

where $\alpha > 0$ is a learnable sharpening coefficient (constrained to be non-negative via a $\mathrm{softplus}$ function in practice). The final sharpened attention is obtained by re-normalizing each row:

$$\hat{A}_{ij} = \frac{\bar{A}_{ij}}{\sum\limits_{k=1}^{N} \bar{A}_{ik} + \varepsilon}. \tag{49}$$

This sharpening operation produces sparse, block-adaptive attention distributions, enabling each token to focus on a compact set of highly relevant intra-block relations while suppressing diffuse or noisy responses.

**Block-wise Application.** The above sharpening operators are independently applied to the four directional cross-block attention matrices $\mathbf{A}_{\text{hist}\rightarrow\text{hist}}$, $\mathbf{A}_{\text{hist}\rightarrow\text{cur}}$, $\mathbf{A}_{\text{cur}\rightarrow\text{hist}}$, and $\mathbf{A}_{\text{cur}\rightarrow\text{cur}}$. This design enables structured and controllable enhancement of dependencies between the historical latent manifold and the current observation manifold. Importantly, the sharpening mechanism preserves the global attention decomposition structure while improving the discriminability and stability of imputation inference.

## D. Extended Theoretical Derivation for Conditional Temporal Inference

This appendix provides a more comprehensive derivation of the probabilistic formulation and expected risk results underlying the Conditional Temporal Inference paradigm presented in the main text. The derivation strictly follows the notation used throughout the paper: $D = (X, M, \mathcal{T})$, $X_{\text{hist}}$, $X_{\text{imp}}$, $X_{\text{imp}}^{(m=0)}$, $X_{\text{imp}}^{(m=1)}$, $\Delta$, as well as the model $f_\theta(\cdot)$ and the conditional distribution $p_\theta(\cdot)$.

### D.1. Masked imputation as conditional risk minimization

Under the masked setting, the imputation model receives supervision only at missing positions. Given a sample $(X, M, \Delta)$, let $\Omega(M)$ denote the set of missing indices. Let $\ell_{\Omega(M)}(\cdot, \cdot)$ denote a masked reconstruction loss, defined only over indices in $\Omega(M)$. For an arbitrary conditioning information set $\mathcal{I}$, we define the conditional expected risk over missing positions as

$$\mathcal{R}(\theta \mid \mathcal{I}) = \mathbb{E}_{X_{\text{imp}}^{(m=1)} \mid \mathcal{I}} \left[ \ell_{\Omega(M)} \left( X_{\text{imp}}^{(m=1)}, f_\theta(\mathcal{I}) \right) \right]. \tag{50}$$

To avoid suppressing details of the conditional distribution, we rewrite (50) in integral form:

$$\mathcal{R}(\theta \mid \mathcal{I}) = \int \ell_{\Omega(M)} \left( x_{\text{imp}}^{(m=1)}, f_\theta(\mathcal{I}) \right) p \left( x_{\text{imp}}^{(m=1)} \mid \mathcal{I} \right) dx_{\text{imp}}^{(m=1)}. \tag{51}$$

Taking expectation further over the data distribution and the missingness mechanism yields the overall risk:

$$\mathcal{J}(\theta) = \mathbb{E}_{X,M,\Delta} \left[ \mathcal{R}(\theta \mid \mathcal{I}) \right] = \int \mathcal{R}(\theta \mid \mathcal{I}) \, p(X, M, \Delta) \, dX \, dM \, d\Delta. \tag{52}$$

These expressions indicate that, under a fixed loss $\ell_{\Omega(M)}(\cdot)$, the attainable levels of both $\mathcal{R}(\theta \mid \mathcal{I})$ and $\mathcal{J}(\theta)$ are governed by the true conditional distribution $p(X_{\text{imp}}^{(m=1)} \mid \mathcal{I})$. Consequently, the essential difference between inference paradigms lies in the choice of the conditioning information set $\mathcal{I}$.

### D.2. Temporal-dependence assumption in a purely probabilistic form

To reflect the inherent temporal structure of time-series data, we introduce a minimal yet crucial probabilistic assumption: even after conditioning on the observed portion within the target window, historical context still provides additional constraints on missing targets. Formally, there exists a subset of samples such that

$$p \left( X_{\text{imp}}^{(m=1)} \mid X_{\text{hist}}, X_{\text{imp}}^{(m=0)}, M, \Delta \right) \neq p \left( X_{\text{imp}}^{(m=1)} \mid X_{\text{imp}}^{(m=0)}, M, \Delta \right). \tag{53}$$

This assumption directly captures the notion that *historical information is informative*, without relying on any latent-variable modeling. When this assumption holds, discarding $X_{\text{hist}}$ alters the conditional distribution of the missing targets, thereby changing the weighting term $p(x_{\text{imp}}^{(m=1)} \mid \mathcal{I})$ in (51).

### D.3. Historical marginalization and the induced mixture inference

We now formalize the probabilistic meaning of not injecting historical information. If $X_{\text{hist}}$ is not explicitly used during inference, the conditioning information sets become

$$\mathcal{I}_{\text{intra}} = \{X_{\text{imp}}^{(m=0)}, M, \Delta\}, \qquad \mathcal{I}_{\text{full}} = \{X_{\text{hist}}, X_{\text{imp}}^{(m=0)}, M, \Delta\}. \tag{54}$$

By the law of total probability (marginalizing over $X_{\text{hist}}$), we obtain

$$p\Big(X_{\text{imp}}^{(m=1)} \mid \mathcal{I}_{\text{intra}}\Big) = \int p\Big(X_{\text{imp}}^{(m=1)}, X_{\text{hist}} \mid \mathcal{I}_{\text{intra}}\Big) \, dX_{\text{hist}} \tag{55}$$

$$= \int p\Big(X_{\text{imp}}^{(m=1)} \mid X_{\text{hist}}, \mathcal{I}_{\text{intra}}\Big) p(X_{\text{hist}} \mid \mathcal{I}_{\text{intra}}) \, dX_{\text{hist}} \tag{56}$$

$$= \int p\Big(X_{\text{imp}}^{(m=1)} \mid \mathcal{I}_{\text{full}}\Big) p\Big(X_{\text{hist}} \mid X_{\text{imp}}^{(m=0)}, M, \Delta\Big) \, dX_{\text{hist}}. \tag{57}$$

Equations (55)–(56) follow from the decomposition of conditional joint distributions, while (57) rewrites the conditioning set using the notation $\mathcal{I}_{\text{full}}$. Equation (57) explicitly shows that *discarding historical information* is equivalent to taking a weighted average over all possible historical trajectories, with weights given by $p(X_{\text{hist}} \mid X_{\text{imp}}^{(m=0)}, M, \Delta)$.

To further expand the weighting term, we apply Bayes' rule to $p(X_{\text{hist}} \mid X_{\text{imp}}^{(m=0)}, M, \Delta)$:

$$p\Big(X_{\text{hist}} \mid X_{\text{imp}}^{(m=0)}, M, \Delta\Big) = \frac{p\Big(X_{\text{imp}}^{(m=0)} \mid X_{\text{hist}}, M, \Delta\Big) p(X_{\text{hist}} \mid M, \Delta)}{p\Big(X_{\text{imp}}^{(m=0)} \mid M, \Delta\Big)} \tag{58}$$

$$= \frac{p\Big(X_{\text{imp}}^{(m=0)} \mid X_{\text{hist}}, M, \Delta\Big) p(X_{\text{hist}} \mid M, \Delta)}{\int p\Big(X_{\text{imp}}^{(m=0)} \mid \tilde{X}_{\text{hist}}, M, \Delta\Big) p\Big(\tilde{X}_{\text{hist}} \mid M, \Delta\Big) \, d\tilde{X}_{\text{hist}}}. \tag{59}$$

As a result, ignoring historical context induces an explicit mixture:

$$p\Big(X_{\text{imp}}^{(m=1)} \mid \mathcal{I}_{\text{intra}}\Big) = \int p\Big(X_{\text{imp}}^{(m=1)} \mid \mathcal{I}_{\text{full}}\Big) \frac{p(X_{\text{imp}}^{(m=0)} \mid X_{\text{hist}}, M, \Delta) p(X_{\text{hist}} \mid M, \Delta)}{\int p(X_{\text{imp}}^{(m=0)} \mid \tilde{X}_{\text{hist}}, M, \Delta) p(\tilde{X}_{\text{hist}} \mid M, \Delta) \, d\tilde{X}_{\text{hist}}} \, dX_{\text{hist}}. \tag{60}$$

Equation (60) reveals the origin of "intra-window similarity dominance": when $X_{\text{imp}}^{(m=0)}$ is sparsely observed under the mask, the weighting term favors historical segments that better explain the observed entries, causing inference to be driven primarily by relative consistency between the observed window and candidate histories, rather than by true temporal evolution constraints.

### D.4. From conditional distributions to expected risk: iterated expectation

We now incorporate the above marginalization–mixture structure into the expected risk formulation. For $\mathcal{I}_{\text{intra}}$, by (51) we have

$$\mathcal{R}(\theta \mid \mathcal{I}_{\text{intra}}) = \int \ell_{\Omega(M)}\Big(x_{\text{imp}}^{(m=1)}, f_\theta(\mathcal{I}_{\text{intra}})\Big) p\Big(x_{\text{imp}}^{(m=1)} \mid \mathcal{I}_{\text{intra}}\Big) \, dx_{\text{imp}}^{(m=1)} \tag{61}$$

$$= \int \ell_{\Omega(M)}\Big(x_{\text{imp}}^{(m=1)}, f_\theta(\mathcal{I}_{\text{intra}})\Big) \left[\int p\Big(x_{\text{imp}}^{(m=1)} \mid \mathcal{I}_{\text{full}}\Big) p(X_{\text{hist}} \mid \mathcal{I}_{\text{intra}}) \, dX_{\text{hist}}\right] dx_{\text{imp}}^{(m=1)} \tag{62}$$

$$= \int \left[\int \ell_{\Omega(M)}\Big(x_{\text{imp}}^{(m=1)}, f_\theta(\mathcal{I}_{\text{intra}})\Big) p\Big(x_{\text{imp}}^{(m=1)} \mid \mathcal{I}_{\text{full}}\Big) \, dx_{\text{imp}}^{(m=1)}\right] p(X_{\text{hist}} \mid \mathcal{I}_{\text{intra}}) \, dX_{\text{hist}}. \tag{63}$$

Equation (62) replaces $p(x_{\text{imp}}^{(m=1)} \mid \mathcal{I}_{\text{intra}})$ using the marginalized form in (56), while (63) exchanges the order of integration. Thus, $\mathcal{R}(\theta \mid \mathcal{I}_{\text{intra}})$ corresponds to optimizing an expected loss in which the influence of historical trajectories on the missing targets is marginalized, effectively attenuating temporal evolution constraints through posterior-weighted averaging.

*Table 5.* Experiments with Joint Supervision at Low-to-Moderate Missing Rates under MCAR

| | | Statistics | | | Time series model | | | Reconstruction Model | | | Diffusion Model | Unified Model | | Proposed | |
| | | MICE | | Missforest | | DeformableTST | | XPatch | | SAITS | | ImputeFormer | | SSSD | | UNiTS | | CBiT | |
| | | MAE | RMSE | MAE | RMSE | MAE | RMSE | MAE | RMSE | MAE | RMSE | MAE | RMSE | MAE | RMSE | MAE | RMSE | MAE | RMSE |
|---|---|---|---|---|---|---|---|---|---|---|---|---|---|---|---|---|---|---|---|
| ETTh1 | 10% | 0.3997 | 0.6770 | 0.3483 | 0.4950 | 0.3829 | 0.5492 | 0.4329 | 0.6118 | 0.2467 | 0.3686 | 0.3715 | 0.5753 | 0.2369 | 0.3331 | 0.3026 | 0.4574 | 0.1658 | 0.2849 |
| | 20% | 0.4633 | 0.7562 | 0.3799 | 0.5583 | 0.4071 | 0.5775 | 0.4704 | 0.6475 | 0.2730 | 0.4534 | 0.4058 | 0.6307 | 0.2374 | 0.3395 | 0.3900 | 0.6039 | 0.175 | 0.2956 |
| | 30% | 0.5403 | 0.8424 | 0.4102 | 0.6141 | 0.4576 | 0.6371 | 0.5048 | 0.6917 | 0.3110 | 0.5138 | 0.4070 | 0.6469 | 0.2986 | 0.4223 | 0.4733 | 0.7029 | 0.1959 | 0.3282 |
| | 40% | 0.5941 | 0.8917 | 0.4515 | 0.6799 | 0.4789 | 0.6507 | 0.5504 | 0.7477 | 0.4024 | 0.6401 | 0.7784 | 1.0210 | 0.3803 | 0.5165 | 0.5707 | 0.8362 | 0.2068 | 0.3411 |
| | 50% | 0.6462 | 0.9388 | 0.4933 | 0.7417 | 0.4299 | 0.6067 | 0.5996 | 0.8078 | 0.3729 | 0.6096 | 0.7756 | 1.0167 | 0.4144 | 0.5709 | 0.6304 | 0.9076 | 0.2217 | 0.367 |
| ETTh2 | 10% | 0.5729 | 0.7336 | 0.2058 | 0.3155 | 0.1879 | 0.2661 | 0.2123 | 0.2911 | 0.1993 | 0.2661 | 0.1883 | 0.2635 | 0.2877 | 0.3765 | 0.1479 | 0.2175 | 0.0999 | 0.1477 |
| | 20% | 0.6010 | 0.7649 | 0.2152 | 0.3311 | 0.1846 | 0.2637 | 0.2303 | 0.3130 | 0.1540 | 0.2048 | 0.1639 | 0.2266 | 0.3165 | 0.4143 | 0.1970 | 0.2787 | 0.1038 | 0.1524 |
| | 30% | 0.6617 | 0.8330 | 0.2270 | 0.3495 | 0.2082 | 0.2889 | 0.2681 | 0.3669 | 0.2038 | 0.2662 | 0.1696 | 0.2330 | 0.4154 | 0.5223 | 0.2192 | 0.3079 | 0.1198 | 0.1798 |
| | 40% | 0.6489 | 0.8332 | 0.2457 | 0.3814 | 0.2390 | 0.3269 | 0.3146 | 0.4310 | 0.2220 | 0.2837 | 0.1932 | 0.2706 | 0.4164 | 0.5692 | 0.2496 | 0.3493 | 0.1255 | 0.1877 |
| | 50% | 0.7101 | 0.9252 | 0.2658 | 0.4163 | 0.2403 | 0.3261 | 0.3570 | 0.4846 | 0.2127 | 0.2824 | 0.2065 | 0.2853 | 0.5283 | 0.6839 | 0.2780 | 0.3849 | 0.1354 | 0.1985 |
| ETTm1 | 10% | 0.3989 | 0.6746 | 0.3451 | 0.4940 | 0.1784 | 0.2873 | 0.2096 | 0.3117 | 0.1074 | 0.1706 | 0.1430 | 0.2245 | 0.1227 | 0.1815 | 0.1768 | 0.2792 | 0.1041 | 0.1835 |
| | 20% | 0.4668 | 0.7551 | 0.3802 | 0.5571 | 0.1846 | 0.2886 | 0.2492 | 0.3609 | 0.1213 | 0.1934 | 0.1601 | 0.2755 | 0.1561 | 0.2286 | 0.2125 | 0.3361 | 0.1041 | 0.1802 |
| | 30% | 0.5410 | 0.8236 | 0.4075 | 0.6029 | 0.2073 | 0.3200 | 0.2998 | 0.4294 | 0.1274 | 0.2137 | 0.1658 | 0.2969 | 0.1545 | 0.2322 | 0.2621 | 0.4096 | 0.1109 | 0.1929 |
| | 40% | 0.5945 | 0.8858 | 0.4403 | 0.6591 | 0.2039 | 0.3072 | 0.3348 | 0.4759 | 0.1466 | 0.2448 | 0.1930 | 0.3457 | 0.1753 | 0.2587 | 0.3238 | 0.4992 | 0.1177 | 0.2119 |
| | 50% | 0.6523 | 0.9049 | 0.4816 | 0.7183 | 0.1956 | 0.2952 | 0.3672 | 0.5114 | 0.1634 | 0.2900 | 0.2128 | 0.3949 | 0.2145 | 0.3049 | 0.3901 | 0.5952 | 0.128 | 0.2275 |
| ETTm2 | 10% | 0.5706 | 0.7360 | 0.2043 | 0.3156 | 0.1043 | 0.1580 | 0.1053 | 0.1656 | 0.1179 | 0.1533 | 0.0702 | 0.1105 | 0.1212 | 0.1835 | 0.1708 | 0.2361 | 0.0639 | 0.1029 |
| | 20% | 0.6186 | 0.7858 | 0.2154 | 0.3352 | 0.1098 | 0.1621 | 0.1204 | 0.1905 | 0.1040 | 0.1380 | 0.0731 | 0.1102 | 0.1812 | 0.2238 | 0.1274 | 0.1995 | 0.0728 | 0.104 |
| | 30% | 0.6647 | 0.8402 | 0.2283 | 0.3512 | 0.1133 | 0.1687 | 0.1694 | 0.2368 | 0.1005 | 0.1339 | 0.0823 | 0.1304 | 0.1143 | 0.1564 | 0.1556 | 0.2447 | 0.0728 | 0.1151 |
| | 40% | 0.6913 | 0.8834 | 0.2401 | 0.3742 | 0.1453 | 0.2297 | 0.2046 | 0.2817 | 0.1161 | 0.1560 | 0.0715 | 0.1137 | 0.1348 | 0.1830 | 0.2192 | 0.3352 | 0.0793 | 0.1246 |
| | 50% | 0.6558 | 0.8508 | 0.2611 | 0.4109 | 0.2075 | 0.3080 | 0.2176 | 0.3033 | 0.1255 | 0.1682 | 0.0780 | 0.1235 | 0.1098 | 0.1516 | 0.2318 | 0.3613 | 0.0922 | 0.142 |
| WTH | 10% | 0.1984 | 0.4356 | 0.2867 | 0.4635 | 0.2218 | 0.4342 | 0.2568 | 0.4482 | 0.1242 | 0.3652 | 0.1390 | 0.3893 | 0.2275 | 0.4268 | 0.2309 | 0.4527 | 0.1165 | 0.3285 |
| | 20% | 0.2213 | 0.4824 | 0.2989 | 0.4817 | 0.2442 | 0.4616 | 0.2836 | 0.4682 | 0.1218 | 0.3595 | 0.1337 | 0.3975 | 0.1775 | 0.3899 | 0.2158 | 0.4378 | 0.1196 | 0.3381 |
| | 30% | 0.3064 | 0.8810 | 0.3201 | 0.5091 | 0.2442 | 0.4519 | 0.3266 | 0.5118 | 0.1362 | 0.3694 | 0.1552 | 0.3880 | 0.2995 | 0.4910 | 0.2482 | 0.4708 | 0.1273 | 0.3423 |
| | 40% | 0.3593 | 1.1316 | 0.3459 | 0.5471 | 0.2583 | 0.4650 | 0.3972 | 0.5859 | 0.1627 | 0.4111 | 0.1578 | 0.4189 | 0.3198 | 0.5203 | 0.3267 | 0.5644 | 0.1463 | 0.3745 |
| | 50% | 0.3833 | 0.8032 | 0.3763 | 0.5882 | 0.2410 | 0.4638 | 0.4014 | 0.6047 | 0.1514 | 0.4292 | 0.1748 | 0.4255 | 0.2760 | 0.4964 | 0.4220 | 0.6756 | 0.1487 | 0.3863 |
| Air | 10% | 0.3182 | 0.4447 | 0.2643 | 0.3969 | 0.2984 | 0.4276 | 0.3154 | 0.4559 | 0.1667 | 0.2547 | 0.2481 | 0.3878 | 0.2245 | 0.3210 | 0.2100 | 0.3140 | 0.129 | 0.2183 |
| | 20% | 0.3668 | 0.5154 | 0.2763 | 0.4132 | 0.3145 | 0.4465 | 0.3629 | 0.5110 | 0.1730 | 0.2754 | 0.2458 | 0.3935 | 0.3054 | 0.4180 | 0.2999 | 0.4435 | 0.1517 | 0.2732 |
| | 30% | 0.4504 | 0.6630 | 0.2840 | 0.4279 | 0.3469 | 0.4897 | 0.3997 | 0.5647 | 0.1806 | 0.2901 | 0.2317 | 0.3703 | 0.3714 | 0.5024 | 0.3710 | 0.5439 | 0.1691 | 0.2976 |
| | 40% | 0.4763 | 0.6874 | 0.3051 | 0.4558 | 0.3611 | 0.5087 | 0.4463 | 0.6316 | 0.1939 | 0.3013 | 0.2353 | 0.3707 | 0.3715 | 0.5158 | 0.4344 | 0.6284 | 0.1938 | 0.3183 |
| | 50% | 0.5115 | 0.7177 | 0.3242 | 0.4810 | 0.3524 | 0.5162 | 0.4659 | 0.6569 | 0.2253 | 0.3439 | 0.2392 | 0.3730 | 0.3635 | 0.5179 | 0.5176 | 0.7188 | 0.2154 | 0.3575 |
| Power | 10% | 0.8711 | 1.0246 | 0.3854 | 0.5830 | 0.0542 | 0.1136 | 0.1015 | 0.1583 | 0.0841 | 0.1271 | 0.0485 | 0.1163 | 0.0590 | 0.0956 | 0.0693 | 0.1537 | 0.0336 | 0.0874 |
| | 20% | 0.8922 | 1.0655 | 0.3953 | 0.5464 | 0.0673 | 0.1205 | 0.1351 | 0.2023 | 0.1048 | 0.1763 | 0.0520 | 0.1174 | 0.0713 | 0.1126 | 0.1107 | 0.2291 | 0.0347 | 0.0901 |
| | 30% | 0.9445 | 1.1449 | 0.4107 | 0.5744 | 0.0775 | 0.1384 | 0.1820 | 0.2628 | 0.1370 | 0.2080 | 0.0682 | 0.1666 | 0.0785 | 0.1274 | 0.1809 | 0.3336 | 0.0397 | 0.1045 |
| | 40% | 1.0005 | 1.2457 | 0.4332 | 0.6115 | 0.2122 | 0.3934 | 0.2492 | 0.3523 | 0.1356 | 0.2334 | 0.0598 | 0.1413 | 0.1383 | 0.2120 | 0.3307 | 0.5313 | 0.0454 | 0.1137 |
| | 50% | 1.0319 | 1.3301 | 0.4542 | 0.6466 | 0.2573 | 0.3990 | 0.3027 | 0.4356 | 0.1862 | 0.2992 | 0.0676 | 0.1560 | 0.1182 | 0.1904 | 0.4264 | 0.6204 | 0.0557 | 0.1502 |
| Solar | 10% | 0.0745 | 0.1634 | 0.0858 | 0.1769 | 0.2437 | 0.3738 | 0.2969 | 0.4608 | 0.1029 | 0.1975 | 0.2007 | 0.3720 | 3.0619 | 3.8361 | 0.1710 | 0.2592 | 0.0621 | 0.1412 |
| | 20% | 0.0908 | 0.1948 | 0.0858 | 0.1774 | 0.2668 | 0.4078 | 0.4361 | 0.4637 | 0.1035 | 0.1959 | 0.1793 | 0.3402 | 3.0618 | 3.8415 | 0.1924 | 0.2961 | 0.0664 | 0.1446 |
| | 30% | 0.0908 | 0.1896 | 0.0870 | 0.1812 | 0.2997 | 0.4543 | 0.4655 | 0.6303 | 0.1104 | 0.2095 | 0.1596 | 0.3101 | 3.0760 | 3.8581 | 0.2307 | 0.3530 | 0.0682 | 0.1491 |
| | 40% | 0.0958 | 0.1913 | 0.0885 | 0.1834 | 0.3479 | 0.5194 | 0.4684 | 0.6491 | 0.1177 | 0.2165 | 0.1568 | 0.2976 | 3.0740 | 3.8553 | 0.3010 | 0.4646 | 0.0718 | 0.1526 |
| | 50% | 0.1016 | 0.1912 | 0.0903 | 0.1878 | 0.3446 | 0.5356 | 0.4784 | 0.6761 | 0.1531 | 0.2489 | 0.1601 | 0.3191 | 3.0858 | 3.8658 | 0.3579 | 0.5094 | 0.0729 | 0.1572 |

For comparison, CTIP formulates imputation as conditional inference given the realized historical trajectory,

$$p\left(X_{\text{imp}}^{(m=1)} \mid \mathcal{I}_{\text{full}}\right) = p\left(X_{\text{imp}}^{(m=1)} \mid X_{\text{hist}}, X_{\text{imp}}^{(m=0)}, M, \Delta\right), \tag{64}$$

and does not marginalize historical information within the conditional distribution. When taking expectation over the distribution of temporal trajectories, the overall expected risk under CTIP can still be written, by the law of total probability, as

$$\mathbb{E}_{X,M,\Delta}[\mathcal{R}(\theta \mid \mathcal{I}_{\text{full}})] = \int \mathcal{R}\left(\theta \mid X_{\text{hist}}, X_{\text{imp}}^{(m=0)}, M, \Delta\right) p(X_{\text{hist}}, X_{\text{imp}}^{(m=0)}, M, \Delta) \, dX_{\text{hist}} \, dX_{\text{imp}}^{(m=0)}. \tag{65}$$

Crucially, the averaging over historical trajectories in the above expression occurs *outside* the conditional inference, whereas intra-window inference marginalizes historical information directly within the conditional distribution. This distinction explains why CTIP preserves history-conditioned temporal constraints during optimization and yields a strictly stronger inference paradigm.

## D.5. Achievable risk inequality under information-set inclusion

We now prove the core inequality stated in the main text: under the same loss function, the minimum achievable conditional expected risk with a richer conditioning information set is no greater than that attainable with a weaker one.

We first define two function classes (policy sets):

$$\mathcal{F}_{\text{intra}} = \left\{g(\cdot) \; : \; \hat{X}_{\text{imp}}^{(m=1)} = g(X_{\text{imp}}^{(m=0)}, M, \Delta)\right\}, \tag{66}$$

$$\mathcal{F}_{\text{full}} = \left\{h(\cdot) \; : \; \hat{X}_{\text{imp}}^{(m=1)} = h(X_{\text{hist}}, X_{\text{imp}}^{(m=0)}, M, \Delta)\right\}. \tag{67}$$

There is an evident inclusion relationship:

$$\mathcal{F}_{\text{intra}} \subseteq \mathcal{F}_{\text{full}}, \tag{68}$$

*Table 6.* Experiments with Missing-Only Supervision at Low-to-Moderate Missing Rates under MCAR.

| | | Time series model | | | | Reconstruction Model | | | | Diffusion Model | | Unified Model | | Proposed | |
| | | DeformableTST | | XPatch | | SAITS | | ImputeFormer | | SSSD | | UNiTS | | CBiT | |
| | | MAE | RMSE | MAE | RMSE | MAE | RMSE | MAE | RMSE | MAE | RMSE | MAE | RMSE | MAE | RMSE |
|---|---|---|---|---|---|---|---|---|---|---|---|---|---|---|---|
| ETTh1 | 10% | 0.4066 | 0.5853 | 0.4504 | 0.6379 | 0.3336 | 0.4672 | 0.4287 | 0.6913 | 0.2821 | 0.5767 | 0.6371 | 0.9591 | 0.2002 | 0.3432 |
| | 20% | 0.3903 | 0.5574 | 0.4690 | 0.6635 | 0.2568 | 0.4325 | 0.3735 | 0.5934 | 0.2362 | 0.4179 | 0.4683 | 0.6909 | 0.1862 | 0.3098 |
| | 30% | 0.3966 | 0.5595 | 0.4946 | 0.6871 | 0.3076 | 0.5219 | 0.7764 | 1.0180 | 0.2923 | 0.4317 | 0.5218 | 0.7588 | 0.2094 | 0.3400 |
| | 40% | 0.4140 | 0.5894 | 0.5151 | 0.7176 | 0.3288 | 0.5498 | 0.7783 | 1.0207 | 0.2724 | 0.4069 | 0.5793 | 0.8268 | 0.2145 | 0.3479 |
| | 50% | 0.4395 | 0.6311 | 0.5583 | 0.7762 | 0.3795 | 0.6210 | 0.7738 | 1.0140 | 0.3673 | 0.5306 | 0.6522 | 0.9112 | 0.2389 | 0.3756 |
| ETTh2 | 10% | 0.1989 | 0.2739 | 0.2128 | 0.2901 | 0.1869 | 0.2507 | 0.2011 | 0.2754 | 0.1781 | 0.4543 | 0.1680 | 0.2415 | 0.1597 | 0.2267 |
| | 20% | 0.1930 | 0.2700 | 0.2268 | 0.3092 | 0.1503 | 0.2054 | 0.4444 | 0.5587 | 0.2096 | 0.3754 | 0.1941 | 0.2747 | 0.1178 | 0.1691 |
| | 30% | 0.1873 | 0.2617 | 0.2354 | 0.3231 | 0.1779 | 0.2342 | 0.1711 | 0.2350 | 0.2086 | 0.3465 | 0.2421 | 0.3414 | 0.1261 | 0.1867 |
| | 40% | 0.1914 | 0.2664 | 0.2670 | 0.3672 | 0.1958 | 0.2703 | 0.5125 | 0.6070 | 0.1416 | 0.2021 | 0.2823 | 0.3975 | 0.1386 | 0.2010 |
| | 50% | 0.1891 | 0.2636 | 0.2987 | 0.4156 | 0.1890 | 0.2619 | 0.5326 | 0.6456 | 0.2507 | 0.3524 | 0.3554 | 0.4853 | 0.1648 | 0.2309 |
| ETTm1 | 10% | 0.1774 | 0.2731 | 0.2081 | 0.3045 | 0.1671 | 0.2415 | 0.1419 | 0.2283 | 0.1431 | 0.2453 | 0.1771 | 0.2751 | 0.1213 | 0.2018 |
| | 20% | 0.1809 | 0.2770 | 0.2319 | 0.3350 | 0.1207 | 0.2067 | 0.1657 | 0.3057 | 0.1711 | 0.2613 | 0.2470 | 0.3935 | 0.1109 | 0.1912 |
| | 30% | 0.1823 | 0.2761 | 0.2512 | 0.3598 | 0.1775 | 0.3166 | 0.7696 | 1.0023 | 0.2139 | 0.3226 | 0.3372 | 0.5431 | 0.1141 | 0.1972 |
| | 40% | 0.1784 | 0.2728 | 0.2733 | 0.3907 | 0.1544 | 0.2631 | 0.7738 | 1.0057 | 0.1963 | 0.2995 | 0.5289 | 0.8186 | 0.1395 | 0.2334 |
| | 50% | 0.1844 | 0.2833 | 0.2666 | 0.3789 | 0.1862 | 0.3043 | 0.6386 | 0.9022 | 0.2210 | 0.3382 | 0.5399 | 0.7715 | 0.1734 | 0.3050 |
| WTH | 10% | 0.2279 | 0.4289 | 0.2605 | 0.4486 | 0.1377 | 0.3541 | 0.1461 | 0.3945 | 0.2507 | 0.6291 | 0.1820 | 0.4034 | 0.1164 | 0.3290 |
| | 20% | 0.2268 | 0.4342 | 0.2810 | 0.4678 | 0.1403 | 0.3639 | 0.1573 | 0.3851 | 0.2636 | 0.6066 | 0.2230 | 0.4374 | 0.1357 | 0.3620 |
| | 30% | 0.2284 | 0.4397 | 0.2931 | 0.4744 | 0.1393 | 0.3765 | 0.1478 | 0.4031 | 0.2264 | 0.4929 | 0.2700 | 0.4857 | 0.1241 | 0.3665 |
| | 40% | 0.2379 | 0.4481 | 0.3166 | 0.5035 | 0.1547 | 0.4038 | 0.1785 | 0.4249 | 0.2524 | 0.4893 | 0.3792 | 0.5903 | 0.1366 | 0.4279 |
| | 50% | 0.2455 | 0.4520 | 0.3316 | 0.5145 | 0.1815 | 0.4269 | 0.7734 | 0.9728 | 0.2624 | 0.5157 | 0.3896 | 0.6154 | 0.1777 | 0.4240 |
| Air | 10% | 0.2918 | 0.4258 | 0.3074 | 0.4446 | 0.1670 | 0.2539 | 0.2614 | 0.3873 | 1.8599 | 2.5977 | 0.2467 | 0.3698 | 0.1687 | 0.2909 |
| | 20% | 0.2993 | 0.4318 | 0.3536 | 0.5047 | 0.2055 | 0.3063 | 0.2495 | 0.3933 | 1.0747 | 1.7335 | 0.2878 | 0.4216 | 0.1621 | 0.2763 |
| | 30% | 0.3095 | 0.4535 | 0.3728 | 0.5350 | 0.2161 | 0.3180 | 0.2268 | 0.3628 | 0.2149 | 0.3474 | 0.3511 | 0.5090 | 0.1735 | 0.2921 |
| | 40% | 0.3290 | 0.4804 | 0.4180 | 0.5914 | 0.2667 | 0.3799 | 0.2425 | 0.3751 | 0.2346 | 0.3711 | 0.4768 | 0.6883 | 0.1934 | 0.3105 |
| | 50% | 0.3415 | 0.5039 | 0.4659 | 0.6609 | 0.2755 | 0.3958 | 0.2756 | 0.4144 | 0.2697 | 0.4051 | 0.5554 | 0.7705 | 0.2151 | 0.3425 |
| Power | 10% | 0.0584 | 0.1088 | 0.1160 | 0.1727 | 0.0758 | 0.1163 | 0.0524 | 0.1215 | 0.1668 | 0.4956 | 0.0733 | 0.1418 | 0.0371 | 0.0891 |
| | 20% | 0.0589 | 0.1078 | 0.1499 | 0.2136 | 0.0878 | 0.1401 | 0.0471 | 0.1168 | 0.0565 | 0.0974 | 0.1108 | 0.2092 | 0.0368 | 0.0939 |
| | 30% | 0.0580 | 0.1089 | 0.1890 | 0.2634 | 0.0870 | 0.1407 | 0.0596 | 0.1434 | 0.0444 | 0.1080 | 0.2835 | 0.4327 | 0.0427 | 0.1054 |
| | 40% | 0.0672 | 0.1203 | 0.2179 | 0.3012 | 0.0919 | 0.1539 | 0.0503 | 0.1263 | 0.0785 | 0.1307 | 0.3733 | 0.5450 | 0.0493 | 0.1149 |
| | 50% | 0.0720 | 0.1313 | 0.2815 | 0.3851 | 0.1832 | 0.2820 | 0.0538 | 0.1350 | 0.1014 | 0.1760 | 0.4525 | 0.6945 | 0.0596 | 0.1414 |
| Solar | 10% | 0.2779 | 0.3981 | 0.3728 | 0.5436 | 0.1103 | 0.1992 | 0.1369 | 0.2609 | 3.1342 | 3.9133 | 0.1966 | 0.2857 | 0.1080 | 0.1957 |
| | 20% | 0.2756 | 0.4082 | 0.4924 | 0.7444 | 0.1353 | 0.2179 | 0.1470 | 0.2726 | 3.0571 | 3.8312 | 0.2561 | 0.3702 | 0.0927 | 0.1775 |
| | 30% | 0.2763 | 0.4138 | 0.5055 | 0.7258 | 0.1841 | 0.2627 | 0.1445 | 0.2729 | 3.0719 | 3.8515 | 0.3633 | 0.5255 | 0.0875 | 0.1722 |
| | 40% | 0.3441 | 0.5099 | 0.5833 | 0.8161 | 0.1147 | 0.2077 | 0.1557 | 0.2854 | 3.0770 | 3.8584 | 0.3978 | 0.5849 | 0.1126 | 0.2040 |
| | 50% | 0.3775 | 0.5700 | 0.5851 | 0.8350 | 0.2506 | 0.3364 | 0.1530 | 0.2832 | 3.0730 | 3.8482 | 0.7882 | 1.1123 | 0.1102 | 0.2106 |

since for any $g \in \mathcal{F}_{\text{intra}}$, one can always construct $h \in \mathcal{F}_{\text{full}}$ that ignores $X_{\text{hist}}$, i.e., $h(X_{\text{hist}}, X_{\text{imp}}^{(m=0)}, M, \Delta) = g(X_{\text{imp}}^{(m=0)}, M, \Delta)$.

We define the minimum achievable conditional expected risks under the two function classes as

$$\mathcal{R}_{\text{intra}}^{\star} = \inf_{g \in \mathcal{F}_{\text{intra}}} \mathbb{E}_{X_{\text{imp}}^{(m=1)} | \mathcal{I}_{\text{intra}}} \left[ \ell_{\Omega(M)} \left( X_{\text{imp}}^{(m=1)}, g(X_{\text{imp}}^{(m=0)}, M, \Delta) \right) \right], \tag{69}$$

$$\mathcal{R}_{\text{full}}^{\star} = \inf_{h \in \mathcal{F}_{\text{full}}} \mathbb{E}_{X_{\text{imp}}^{(m=1)} | \mathcal{I}_{\text{full}}} \left[ \ell_{\Omega(M)} \left( X_{\text{imp}}^{(m=1)}, h(X_{\text{hist}}, X_{\text{imp}}^{(m=0)}, M, \Delta) \right) \right]. \tag{70}$$

From the inclusion (68), it immediately follows that

$$\mathcal{R}_{\text{full}}^{\star} \leq \mathcal{R}_{\text{intra}}^{\star}. \tag{71}$$

This result does not depend on the specific form of $\ell_{\Omega(M)}(\cdot)$, but solely on the fact that a richer conditioning set induces a larger class of admissible strategies.

Finally, substituting (71) into the overall risk definition (52) yields the population-level inequality:

$$\mathbb{E}_{X, M, \Delta}[\mathcal{R}_{\text{full}}^{\star}] \leq \mathbb{E}_{X, M, \Delta}[\mathcal{R}_{\text{intra}}^{\star}]. \tag{72}$$

# E. Additional Experimental Results

We further conduct additional experiments to provide a more comprehensive evaluation of the proposed method. These experiments include extended results beyond those reported in the main text, as well as systematic ablation studies, which are detailed in the following sections. In particular, under the MCAR and MAR missing mechanism, we report experimental results for both joint supervision and missing-only supervision across different missing regimes, including low-to-moderate missing rates and high missing rates. The corresponding results are presented and analyzed in detail below.

*Table 7.* Experiments with Joint Supervision at High Missing Rates under MCAR

| | | Statistics | | | | Time series model | | | | Reconstruction Model | | | | Diffusion Model | | Unified Model | | Proposed | |
| | | MICE | | Missforest | | DeformableTST | | XPatch | | SAITS | | ImputeFormer | | SSSD | | UNiTS | | CBiT | |
| | | MAE | RMSE | MAE | RMSE | MAE | RMSE | MAE | RMSE | MAE | RMSE | MAE | RMSE | MAE | RMSE | MAE | RMSE | MAE | RMSE |
|---|---|---|---|---|---|---|---|---|---|---|---|---|---|---|---|---|---|---|---|
| ETTh1 | 60% | 0.7370 | 1.0260 | 0.5375 | 0.8056 | 0.4506 | 0.6642 | 0.6031 | 0.8142 | 0.4556 | 0.7296 | 0.7764 | 1.0175 | 0.4356 | 0.6075 | 0.6188 | 0.8963 | 0.2585 | 0.4054 |
| | 70% | 0.7446 | 1.0033 | 0.5796 | 0.8409 | 0.6315 | 0.8481 | 0.5975 | 0.8215 | 0.5123 | 0.8015 | 0.7714 | 1.0094 | 0.5452 | 0.7203 | 0.7161 | 0.9765 | 0.2888 | 0.4552 |
| | 80% | 0.7882 | 1.0434 | 0.6442 | 0.9278 | 0.5019 | 0.7466 | 0.5238 | 0.8288 | 0.5238 | 0.8216 | 0.7706 | 1.0096 | 0.6944 | 0.9152 | 0.7561 | 1.0164 | 0.3658 | 0.5782 |
| | 90% | 0.8420 | 1.1048 | 0.7099 | 0.9931 | 0.7227 | 0.9586 | 0.6314 | 0.8978 | 0.6385 | 0.9338 | 0.7679 | 1.0058 | 0.8233 | 1.0966 | 0.7367 | 0.9791 | 0.4199 | 0.6498 |
| ETTm1 | 60% | 0.7110 | 0.9901 | 0.5526 | 0.7744 | 0.1897 | 0.2951 | 0.3345 | 0.4690 | 0.1854 | 0.3495 | 0.2250 | 0.3734 | 0.2001 | 0.2977 | 0.5730 | 0.8268 | 0.1464 | 0.2578 |
| | 70% | 0.7364 | 0.9958 | 0.5727 | 0.8350 | 0.1967 | 0.3138 | 0.3303 | 0.4753 | 0.2193 | 0.3767 | 0.7695 | 1.0038 | 0.3150 | 0.4378 | 0.6672 | 0.8989 | 0.1717 | 0.2996 |
| | 80% | 0.7793 | 1.0373 | 0.6388 | 0.9119 | 0.2379 | 0.3744 | 0.3483 | 0.5169 | 0.2895 | 0.5161 | 0.3549 | 0.6453 | 0.3677 | 0.5171 | 0.6505 | 0.9030 | 0.2037 | 0.3509 |
| | 90% | 0.8077 | 1.0429 | 0.7346 | 0.9763 | 0.3352 | 0.5334 | 0.4012 | 0.7064 | 0.4186 | 0.6924 | 0.7710 | 1.0055 | 0.7362 | 0.9796 | 0.6767 | 0.9358 | 0.3369 | 0.5272 |
| WTH | 60% | 0.4384 | 0.7028 | 0.4181 | 0.6466 | 0.2649 | 0.4817 | 0.3747 | 0.5702 | 0.1811 | 0.4324 | 0.1970 | 0.4519 | 0.5650 | 0.7690 | 0.5009 | 0.7196 | 0.1689 | 0.4020 |
| | 70% | 0.6375 | 0.8506 | 0.4738 | 0.7130 | 0.2787 | 0.5089 | 0.3656 | 0.5696 | 0.2007 | 0.4658 | 0.2313 | 0.4723 | 0.6747 | 0.9073 | 0.5950 | 0.7853 | 0.1856 | 0.4377 |
| | 80% | 0.7196 | 0.9266 | 0.5661 | 0.8217 | 0.3085 | 0.5300 | 0.3845 | 0.5957 | 0.2379 | 0.5030 | 0.2849 | 0.5358 | 0.7468 | 0.9882 | 0.6716 | 0.8713 | 0.2194 | 0.4810 |
| | 90% | 0.7476 | 0.9490 | 0.6957 | 0.9663 | 0.3728 | 0.6082 | 0.4197 | 0.6400 | 0.2791 | 0.5316 | 0.3313 | 0.5749 | 0.8132 | 1.0622 | 0.7241 | 0.9165 | 0.2831 | 0.5439 |
| Air | 60% | 0.5756 | 0.7820 | 0.3514 | 0.5204 | 0.3588 | 0.5330 | 0.5063 | 0.7223 | 0.2812 | 0.4088 | 0.2551 | 0.3996 | 0.4624 | 0.6135 | 0.6958 | 0.9388 | 0.2391 | 0.3833 |
| | 70% | 0.6905 | 0.9055 | 0.3832 | 0.5654 | 0.3976 | 0.5805 | 0.5308 | 0.7556 | 0.3097 | 0.4527 | 0.3199 | 0.4739 | 0.5197 | 0.7057 | 0.7600 | 1.0147 | 0.2749 | 0.4256 |
| | 80% | 0.8025 | 1.0588 | 0.4382 | 0.6431 | 0.4403 | 0.6392 | 0.5473 | 0.7654 | 0.3401 | 0.4863 | 0.3866 | 0.5625 | 0.6436 | 0.8566 | 0.7368 | 0.9810 | 0.3284 | 0.5022 |
| | 90% | 0.8390 | 1.1046 | 0.5859 | 0.8282 | 0.5459 | 0.7641 | 0.6036 | 0.8304 | 0.4365 | 0.6156 | 0.4915 | 0.7073 | 0.8191 | 1.0745 | 0.7644 | 1.0125 | 0.4536 | 0.6689 |
| Solar | 60% | 0.1130 | 0.2002 | 0.0932 | 0.1934 | 0.8375 | 1.1401 | 0.9028 | 1.1082 | 0.1416 | 0.2440 | 0.2004 | 0.3365 | 3.0773 | 3.8554 | 0.3785 | 0.5819 | 0.0756 | 0.1596 |
| | 70% | 0.1310 | 0.2187 | 0.0976 | 0.2020 | 0.9775 | 1.2842 | 0.9000 | 1.1112 | 0.1625 | 0.2754 | 0.2064 | 0.3714 | 3.0802 | 3.8607 | 0.4634 | 0.6911 | 0.0907 | 0.1813 |
| | 80% | 0.1897 | 0.2846 | 0.1062 | 0.2193 | 0.9767 | 1.2203 | 0.8708 | 1.0392 | 0.1684 | 0.2783 | 0.2673 | 0.4173 | 3.0812 | 3.8604 | 0.6437 | 0.9533 | 0.1125 | 0.2128 |
| | 90% | 0.5865 | 0.7486 | 0.1255 | 0.2582 | 0.9912 | 1.2651 | 0.8422 | 1.0124 | 0.2074 | 0.3415 | 0.3725 | 0.5614 | 3.0727 | 3.8468 | 0.7375 | 1.0831 | 0.1220 | 0.2242 |

*Table 8.* Experiments with Missing-Only Supervision at High Missing Rates under MCAR.

| | | Time series model | | | | Reconstruction Model | | | | Diffusion Model | | Unified Model | | Proposed | |
| | | DeformableTST | | XPatch | | SAITS | | ImputeFormer | | SSSD | | UNiTS | | CBiT | |
| | | MAE | RMSE | MAE | RMSE | MAE | RMSE | MAE | RMSE | MAE | RMSE | MAE | RMSE | MAE | RMSE |
|---|---|---|---|---|---|---|---|---|---|---|---|---|---|---|---|
| ETTh1 | 60% | 0.4649 | 0.6838 | 0.6070 | 0.8215 | 0.4500 | 0.7147 | 0.7761 | 1.0169 | 0.4399 | 0.6172 | 0.6693 | 0.9382 | 0.2699 | 0.4115 |
| | 70% | 0.4980 | 0.7288 | 0.6035 | 0.8430 | 0.4849 | 0.7578 | 0.7712 | 1.0091 | 0.3972 | 0.5735 | 0.6944 | 0.9345 | 0.2993 | 0.4643 |
| | 80% | 0.5309 | 0.7715 | 0.6181 | 0.8719 | 0.5547 | 0.8479 | 0.7699 | 1.0085 | 0.5548 | 0.7800 | 0.7112 | 0.9442 | 0.4623 | 0.6692 |
| | 90% | 0.6124 | 0.8785 | 0.6369 | 0.9053 | 0.7019 | 1.0216 | 0.7678 | 1.0057 | 0.7020 | 0.9306 | 0.7428 | 0.9901 | 0.4951 | 0.7441 |
| ETTm1 | 60% | 0.1982 | 0.3034 | 0.3699 | 0.5175 | 0.2203 | 0.3628 | 0.7570 | 0.9858 | 0.2502 | 0.3735 | 0.5807 | 0.8611 | 0.1507 | 0.2563 |
| | 70% | 0.2094 | 0.3261 | 0.3432 | 0.4944 | 0.2425 | 0.4264 | 0.2712 | 0.4591 | 0.3109 | 0.4613 | 0.6238 | 0.9168 | 0.1751 | 0.2912 |
| | 80% | 0.2514 | 0.3897 | 0.3771 | 0.5608 | 0.3297 | 0.5703 | 0.6433 | 0.8631 | 0.4123 | 0.5923 | 0.6515 | 0.9564 | 0.2251 | 0.3562 |
| | 90% | 0.6419 | 0.8977 | 0.4652 | 0.7087 | 0.4613 | 0.7459 | 0.7721 | 1.0067 | 0.5748 | 0.7961 | 0.6820 | 0.9391 | 0.2875 | 0.3998 |
| WTH | 60% | 0.2506 | 0.4691 | 0.3567 | 0.5481 | 0.2177 | 0.4687 | 0.7636 | 0.9648 | 0.4231 | 0.6834 | 0.5700 | 0.7457 | 0.1800 | 0.4283 |
| | 70% | 0.2795 | 0.4901 | 0.3746 | 0.5738 | 0.2304 | 0.4918 | 0.7608 | 0.9629 | 0.3683 | 0.6144 | 0.6130 | 0.7903 | 0.1911 | 0.4469 |
| | 80% | 0.3103 | 0.5362 | 0.3923 | 0.6042 | 0.2649 | 0.5134 | 0.7678 | 0.9687 | 0.4090 | 0.6490 | 0.6563 | 0.8467 | 0.2202 | 0.4761 |
| | 90% | 0.3957 | 0.6140 | 0.4660 | 0.6935 | 0.3202 | 0.5521 | 0.7598 | 0.9621 | 0.6398 | 0.8778 | 0.7197 | 0.9189 | 0.7530 | 0.9733 |
| Air | 60% | 0.3902 | 0.5613 | 0.5169 | 0.7216 | 0.2831 | 0.4136 | 0.2877 | 0.4329 | 0.3988 | 0.5728 | 0.6022 | 0.8217 | 0.2385 | 0.3781 |
| | 70% | 0.4182 | 0.6030 | 0.5089 | 0.7116 | 0.3501 | 0.4944 | 0.7835 | 1.0410 | 0.3548 | 0.5066 | 0.6438 | 0.8767 | 0.2832 | 0.4381 |
| | 80% | 0.4658 | 0.6671 | 0.5350 | 0.7630 | 0.3925 | 0.5608 | 0.7851 | 1.0430 | 0.5151 | 0.7204 | 0.7199 | 0.9655 | 0.3452 | 0.5124 |
| | 90% | 0.5671 | 0.8016 | 0.5799 | 0.8005 | 0.4700 | 0.6582 | 0.7850 | 1.0433 | 0.6788 | 0.9244 | 0.7728 | 1.0150 | 0.4666 | 0.6767 |
| Solar | 60% | 0.4708 | 0.6920 | 0.6432 | 0.9058 | 0.1431 | 0.2471 | 0.1713 | 0.3128 | 3.0825 | 3.8614 | 0.3743 | 0.5770 | 0.0977 | 0.1854 |
| | 70% | 0.5316 | 0.7851 | 0.6874 | 0.9665 | 0.1581 | 0.2651 | 0.2059 | 0.3573 | 3.0761 | 3.8517 | 0.4669 | 0.7029 | 0.1325 | 0.2397 |
| | 80% | 0.6393 | 0.9036 | 0.7425 | 1.0292 | 0.1723 | 0.2907 | 0.2328 | 0.4213 | 3.0710 | 3.8494 | 0.7285 | 1.0680 | 0.1226 | 0.2229 |
| | 90% | 0.7292 | 1.0213 | 0.7670 | 0.9979 | 0.1928 | 0.3135 | 0.6876 | 1.1390 | 3.0795 | 3.8599 | 0.7320 | 1.0235 | 0.1231 | 0.2254 |

## E.1. Results under MCAR Missing Mechanism

### E.1.1. JOINT SUPERVISION WITH LOW-TO-MODERATE MISSING RATES

Table 5 reports the imputation performance under joint supervision with low-to-moderate missing rates. Experiments are conducted on a collection of public benchmark datasets, including ETTh1, ETTh2, ETTm1, ETTm2, WTH, Power, Air and Solar, under the MCAR missing mechanism where missing values are generated independently of the underlying observations. During training, joint supervision is adopted by optimizing reconstruction losses on both observed and missing positions. The missing rates are varied within a low-to-moderate range of 10%–50% to evaluate model behavior under partially observed conditions. As shown in Table 5, CBiT demonstrates favorable imputation performance under joint supervision with low-to-moderate missing rates. Across most datasets and missing rate settings, the results indicate that CBiT effectively captures temporal dependencies in partially observed time series, leading to stable and competitive Masked MAE and Masked RMSE values.

### E.1.2. MISSING-ONLY SUPERVISION WITH LOW-TO-MODERATE MISSING RATES

Building upon the joint supervision setting, we further evaluate imputation performance under missing-only supervision with low-to-moderate missing rates. In this setting, training signals are provided exclusively at missing positions, which reflects a more restrictive and practically relevant supervision regime. Experiments are conducted on multiple public benchmark

datasets, including ETTh1, WTH, Power, Air and Solar, under the MCAR missing mechanism, with missing rates ranging from low to moderate levels, spanning 10%–50%. This setting allows us to assess the robustness of different methods when learning is driven solely by incomplete observations. The corresponding results are reported in Table 6.

### E.1.3. JOINT SUPERVISION WITH HIGH MISSING RATES

We further examine the performance of different imputation methods under joint supervision at high missing rates. Experiments are conducted on multiple public benchmark datasets, including ETTh1, ETTm1, WTH, Air and Solar, under the MCAR missing mechanism, with the missing rate increased to a high range (e.g., 60%–90%), corresponding to more challenging imputation conditions. In this regime, the available observations become increasingly sparse, which poses greater difficulty for reliably modeling temporal dependencies. The corresponding results are reported in Table 7.

### E.1.4. MISSING-ONLY SUPERVISION WITH HIGH MISSING RATES

We evaluate the performance of different imputation methods under missing-only supervision at high missing rates. In this setting, training signals are provided exclusively at positions indicated by the missingness mask, while a large proportion of observations are unobserved, resulting in a highly restrictive supervision regime. Experiments are conducted on multiple public benchmark datasets, including ETTh1, ETTm1, WTH, Air and Solar, under the MCAR missing mechanism, with missing rates ranging from 60% to 90%. This setting imposes substantial challenges for learning temporal dependencies from severely incomplete observations. The corresponding results are reported in Table 8.

*Table 9.* Experiments with Joint Supervision under MAR

| | | Statics | | | Time series model | | | | Reconstruction Model | | | | Diffusion Model | | Unified Model | | Proposed | |
|---|---|---|---|---|---|---|---|---|---|---|---|---|---|---|---|---|---|---|---|
| | | MICE | | Missforest | | DeformableTST | | XPatch | | SAITS | | ImputeFormer | | SSSD | | UNiTS | | CBiT | |
| | | MAE | RMSE | MAE | RMSE | MAE | RMSE | MAE | RMSE | MAE | RMSE | MAE | RMSE | MAE | RMSE | MAE | RMSE | MAE | RMSE |
| ETTh1 | 10% | 0.0944 | 0.1455 | 0.5229 | 0.6208 | 0.6428 | 0.8159 | 0.6069 | 0.7515 | 0.0995 | 0.1451 | 0.1327 | 0.1854 | 0.1596 | 0.2204 | 0.2416 | 0.3073 | 0.0963 | 0.1411 |
| | 20% | 0.0977 | 0.1528 | 0.5859 | 0.7087 | 0.6635 | 0.8367 | 0.6527 | 0.8197 | 0.1045 | 0.1567 | 0.1313 | 0.1859 | 0.1444 | 0.1954 | 0.2690 | 0.3500 | 0.1043 | 0.1539 |
| | 30% | 0.0982 | 0.1562 | 0.5907 | 0.7178 | 0.6764 | 0.8617 | 0.7485 | 0.9282 | 0.1065 | 0.1559 | 0.1417 | 0.1993 | 0.1339 | 0.1841 | 0.2936 | 0.3857 | 0.1191 | 0.1792 |
| | 40% | 0.0983 | 0.1562 | 0.6041 | 0.7374 | 0.7083 | 0.9175 | 0.6824 | 0.8716 | 0.1178 | 0.1739 | 0.1403 | 0.1976 | 0.1954 | 0.2470 | 0.3518 | 0.4738 | 0.1049 | 0.1538 |
| | 50% | 0.0995 | 0.1590 | 0.6408 | 0.7916 | 0.7812 | 1.0209 | 0.7812 | 1.0041 | 0.1043 | 0.1579 | 0.1384 | 0.2022 | 0.1439 | 0.1975 | 0.4213 | 0.5834 | 0.1194 | 0.1697 |
| WTH | 10% | 0.1130 | 0.2207 | 0.3703 | 0.4467 | 0.2479 | 0.3861 | 0.3710 | 0.5195 | 0.0441 | 0.2494 | 1.1015 | 1.0145 | 0.0891 | 0.2633 | 0.1716 | 0.3324 | 0.0395 | 0.2425 |
| | 20% | 0.1215 | 0.2352 | 0.4276 | 0.5126 | 0.2543 | 0.4179 | 0.3647 | 0.5200 | 0.0576 | 0.2662 | 0.9902 | 1.1093 | 0.0830 | 0.2585 | 0.1936 | 0.3874 | 0.0560 | 0.2593 |
| | 30% | 0.1256 | 0.2457 | 0.4632 | 0.5544 | 0.6991 | 0.8569 | 0.3913 | 0.5614 | 0.0437 | 0.2392 | 0.9185 | 1.0527 | 0.1224 | 0.2667 | 0.1896 | 0.3927 | 0.0595 | 0.2390 |
| | 40% | 0.1302 | 0.2685 | 0.5081 | 0.6105 | 0.3439 | 0.5869 | 0.4389 | 0.6515 | 0.0539 | 0.2442 | 0.6456 | 0.7731 | 0.2892 | 0.4117 | 0.2102 | 0.4038 | 0.0531 | 0.2569 |
| | 50% | 0.1322 | 0.2742 | 0.5490 | 0.6666 | 0.3707 | 0.6148 | 0.7701 | 0.9512 | 0.0485 | 0.2608 | 0.7778 | 0.9245 | 0.0670 | 0.2484 | 0.2268 | 0.4130 | 0.0457 | 0.2437 |
| Air | 10% | 0.3693 | 0.4864 | 0.6131 | 0.8240 | 0.8188 | 1.0651 | 0.8057 | 1.0739 | 0.3047 | 0.3867 | 0.3181 | 0.4263 | 0.3323 | 0.4427 | 0.4427 | 0.5699 | 0.2671 | 0.3617 |
| | 20% | 0.3537 | 0.4703 | 0.5575 | 0.7706 | 0.7249 | 0.9443 | 0.7429 | 1.0187 | 0.2700 | 0.3853 | 0.2828 | 0.3931 | 0.4559 | 0.6019 | 0.6019 | 0.5328 | 0.2445 | 0.3358 |
| | 30% | 0.3513 | 0.4677 | 0.5708 | 0.7913 | 0.7606 | 1.0390 | 0.7889 | 1.0590 | 0.2672 | 0.3799 | 0.3078 | 0.4089 | 0.3501 | 0.4373 | 0.4373 | 0.4881 | 0.2235 | 0.3067 |
| | 40% | 0.3473 | 0.4636 | 0.5389 | 0.7492 | 0.7086 | 0.9682 | 0.7767 | 1.0545 | 0.3088 | 0.4330 | 0.3069 | 0.4085 | 0.3366 | 0.4447 | 0.4447 | 0.4770 | 0.2347 | 0.3212 |
| | 50% | 0.3472 | 0.4620 | 0.5443 | 0.7759 | 0.7257 | 1.0068 | 0.7394 | 1.0256 | 0.2925 | 0.4210 | 0.2699 | 0.3729 | 0.3022 | 0.3947 | 0.3947 | 0.5081 | 0.2490 | 0.3454 |
| Power | 10% | 1.2312 | 1.4229 | 0.4569 | 0.5767 | 0.3682 | 0.4780 | 0.4046 | 0.5395 | 0.2040 | 0.2771 | 0.1990 | 0.2835 | 0.2839 | 0.4245 | 0.2886 | 0.3847 | 0.1584 | 0.2708 |
| | 20% | 1.2865 | 1.4951 | 0.4726 | 0.5916 | 0.4423 | 0.5643 | 0.4696 | 0.6561 | 0.2171 | 0.3043 | 0.3062 | 0.4648 | 0.2603 | 0.3521 | 0.3516 | 0.4643 | 0.2035 | 0.3200 |
| | 30% | 1.3168 | 1.5380 | 0.4723 | 0.5961 | 0.4733 | 0.6271 | 0.4820 | 0.6114 | 0.2632 | 0.3620 | 0.3934 | 0.5681 | 0.3343 | 0.4738 | 0.3986 | 0.5207 | 0.2339 | 0.3609 |
| | 40% | 1.3278 | 1.5544 | 0.4943 | 0.6233 | 0.5262 | 0.7080 | 0.5273 | 0.7361 | 0.2660 | 0.3449 | 0.3242 | 0.4442 | 0.2894 | 0.4086 | 0.4268 | 0.5591 | 0.2751 | 0.4341 |
| | 50% | 1.3387 | 1.5675 | 0.5069 | 0.6371 | 0.6195 | 0.7798 | 0.5462 | 0.7711 | 0.4806 | 0.6488 | 0.3549 | 0.4638 | 0.3659 | 0.5235 | 0.4454 | 0.5821 | 0.2801 | 0.4644 |
| Solar | 10% | 0.1875 | 0.2714 | 0.3781 | 0.4688 | 0.8753 | 1.0275 | 0.8950 | 1.0349 | 0.2179 | 0.3060 | 0.2780 | 0.3837 | 3.2744 | 4.0898 | 0.3733 | 0.4756 | 0.1674 | 0.2239 |
| | 20% | 0.1881 | 0.2764 | 0.3890 | 0.4943 | 0.8135 | 0.9834 | 0.9925 | 1.1427 | 0.2068 | 0.2849 | 0.2834 | 0.3930 | 3.1831 | 3.9824 | 0.4049 | 0.5119 | 0.1682 | 0.2653 |
| | 30% | 0.1936 | 0.2886 | 0.3988 | 0.5137 | 0.7700 | 0.9504 | 0.8939 | 1.0539 | 0.1866 | 0.2687 | 0.2776 | 0.3942 | 3.1894 | 3.9778 | 0.3461 | 0.4529 | 0.1749 | 0.2649 |
| | 40% | 0.1886 | 0.2871 | 0.4046 | 0.5408 | 0.6486 | 0.8030 | 0.9092 | 1.0818 | 0.1931 | 0.2783 | 0.2931 | 0.3947 | 3.0848 | 3.8918 | 0.3450 | 0.4543 | 0.1584 | 0.2485 |
| | 50% | 0.1896 | 0.2875 | 0.3969 | 0.5481 | 0.7717 | 0.9618 | 0.8813 | 1.0578 | 0.2012 | 0.2884 | 0.2554 | 0.3764 | 3.1581 | 3.9679 | 0.3382 | 0.4368 | 0.1591 | 0.2463 |

## E.2. Results under MAR Missing Mechanism

To assess the performance of CBiT under alternative missing mechanisms, we evaluate the model under the MAR setting using two supervision regimes joint supervision and missing-only supervision.

### E.2.1. JOINT SUPERVISION

Under the MAR missing mechanism, we conduct experiments using joint supervision to evaluate imputation performance. The evaluation is carried out on a suite of public benchmark datasets, including ETTh1, WTH, Power, Air and Solar. The missing rate is varied from 10% to 50% to cover a wide range of incomplete observation scenarios. Results under this setting are summarized in Table 9. Overall, the results indicate that CBiT maintains strong imputation performance under the MAR mechanism.

### E.2.2. MISSING-ONLY SUPERVISION

Under the MAR missing mechanism, we evaluate imputation performance using missing-only supervision. The experiments are conducted on a suite of public benchmark datasets, including Air and Solar, with missing rates ranging from 10% to 50%. This setting characterizes scenarios where supervision is restricted to masked positions. The corresponding results are reported in Table 10. Overall, the results indicate that CBiT remains effective under this more constrained supervision regime.

*Table 10.* Experiments with Missing-Only Supervision under MAR

| | | Time series model | | | | Reconstruction Model | | | | Diffusion Model | | Unified Model | | Proposed | |
| | | DeformableTST | | XPatch | | SAITS | | ImputeFormer | | SSSD | | UNiTS | | CBiT | |
| | | MAE | RMSE | MAE | RMSE | MAE | RMSE | MAE | RMSE | MAE | RMSE | MAE | RMSE | MAE | RMSE |
|---|---|---|---|---|---|---|---|---|---|---|---|---|---|---|---|
| Air | 10% | 0.7441 | 0.9695 | 0.7250 | 0.9497 | 0.2802 | 0.3619 | 0.8528 | 1.0405 | 0.5258 | 0.6850 | 0.5087 | 0.6829 | 0.3331 | 0.4410 |
| | 20% | 0.7348 | 0.9636 | 0.7414 | 0.9802 | 0.3983 | 0.5011 | 0.4661 | 0.5721 | 0.3545 | 0.4911 | 0.4217 | 0.5785 | 0.2746 | 0.3655 |
| | 30% | 0.7072 | 0.9295 | 0.8488 | 1.1095 | 0.3661 | 0.4675 | 0.8829 | 1.1001 | 0.3827 | 0.5215 | 0.4410 | 0.6098 | 0.3205 | 0.4094 |
| | 40% | 0.6544 | 0.8557 | 0.8022 | 1.0553 | 0.2930 | 0.3831 | 0.3635 | 0.4666 | 0.3664 | 0.5202 | 0.4356 | 0.5892 | 0.2805 | 0.3748 |
| | 50% | 0.6559 | 0.8638 | 0.7945 | 1.0425 | 0.2754 | 0.3601 | 0.3241 | 0.4279 | 0.3481 | 0.4955 | 0.4611 | 0.6200 | 0.2417 | 0.3269 |
| Solar | 10% | 0.8335 | 0.9803 | 0.4183 | 0.6045 | 0.2139 | 0.2905 | 0.2729 | 0.3811 | 3.2972 | 4.0989 | 0.5582 | 0.6599 | 0.2033 | 0.2990 |
| | 20% | 0.7843 | 0.9385 | 0.4774 | 0.6956 | 0.2206 | 0.2965 | 0.2501 | 0.3639 | 3.1719 | 3.9806 | 0.5798 | 0.6895 | 0.1969 | 0.2881 |
| | 30% | 0.8163 | 0.9729 | 0.5231 | 0.7409 | 0.2043 | 0.2842 | 0.2491 | 0.3695 | 3.2304 | 4.0095 | 0.5635 | 0.6736 | 0.1698 | 0.2588 |
| | 40% | 0.7521 | 0.9070 | 0.8687 | 1.0256 | 0.2130 | 0.2887 | 0.2385 | 0.3613 | 3.1454 | 3.9278 | 0.4541 | 0.5638 | 0.1785 | 0.2652 |
| | 50% | 0.7405 | 0.8945 | 0.8453 | 1.0100 | 0.1907 | 0.2717 | 0.2372 | 0.3564 | 3.1646 | 3.9221 | 0.4720 | 0.5827 | 0.1693 | 0.2558 |

### E.3. Ablation Analysis

To evaluate the impact of different model configurations on time-series imputation performance, we conduct a systematic ablation study on CBiT. Specifically, $\text{CBiT}_{\text{full}}$ denotes a variant that performs imputation using only the complete global modeling pathway without the decomposed sub-block aggregation, while $\text{CBiT}_{\text{fused}}$ represents a variant that relies solely on the aggregation of decomposed sub-blocks without the full global modeling structure. All models are trained and evaluated under identical settings on the ETTh1, Air and WTH datasets, considering both moderate and high missing rates ranging from 10% to 90% under the MCAR mechanism. The results are summarized in Table 11.

From the results, we observe that both simplified variants exhibit varying degrees of performance degradation across datasets and missing rates, whereas the complete CBiT model achieves the best or near-best performance in most settings and shows stronger overall robustness across datasets and missing rates. These findings indicate that the overall structural design of CBiT is necessary for effective time-series imputation across a wide range of missing scenarios.

*Table 11.* Ablation study of different CBiT modules.

| | | *Missing ratio* | 10% | 20% | 30% | 40% | 50% | 60% | 70% | 80% | 90% |
|---|---|---|---|---|---|---|---|---|---|---|---|---|
| ETTh1 | $\text{CBiT}_{\text{full}}$ | MAE | 0.1679 | 0.1966 | 0.1979 | 0.2070 | 0.2270 | 0.2695 | 0.2987 | 0.3871 | 0.4993 |
| | | RMSE | 0.2795 | 0.3139 | 0.3252 | 0.3432 | 0.3686 | 0.4192 | 0.4682 | 0.6044 | 0.7588 |
| | $\text{CBiT}_{\text{fused}}$ | MAE | 0.1816 | 0.1784 | 0.1943 | 0.2241 | 0.2367 | 0.2655 | 0.3192 | 0.3663 | 0.4890 |
| | | RMSE | 0.2920 | 0.2950 | 0.3244 | 0.3529 | 0.3817 | 0.4132 | 0.4881 | 0.5638 | 0.7366 |
| | CBiT | MAE | 0.1658 | 0.175 | 0.1959 | 0.2068 | 0.2217 | 0.2585 | 0.2888 | 0.3658 | 0.4199 |
| | | RMSE | 0.2849 | 0.2956 | 0.3282 | 0.3411 | 0.3670 | 0.4054 | 0.4552 | 0.5782 | 0.6498 |
| WTH | $\text{CBiT}_{\text{full}}$ | MAE | 0.1206 | 0.1287 | 0.1483 | 0.1477 | 0.1854 | 0.2273 | 0.2369 | 0.2452 | 0.2836 |
| | | RMSE | 0.3440 | 0.3443 | 0.3813 | 0.4022 | 0.4335 | 0.4881 | 0.4867 | 0.5076 | 0.5422 |
| | $\text{CBiT}_{\text{fused}}$ | MAE | 0.1229 | 0.1272 | 0.1364 | 0.1802 | 0.1612 | 0.1728 | 0.1884 | 0.2247 | 0.2957 |
| | | RMSE | 0.3331 | 0.3383 | 0.3602 | 0.4337 | 0.4055 | 0.4184 | 0.4397 | 0.4835 | 0.5548 |
| | CBiT | MAE | 0.1165 | 0.1196 | 0.1273 | 0.1463 | 0.1487 | 0.1689 | 0.1856 | 0.2194 | 0.2831 |
| | | RMSE | 0.3285 | 0.3381 | 0.3423 | 0.3745 | 0.3863 | 0.4020 | 0.4377 | 0.4810 | 0.5439 |
| Air | $\text{CBiT}_{\text{full}}$ | MAE | 0.1466 | 0.1544 | 0.1740 | 0.2009 | 0.2586 | 0.2553 | 0.2749 | 0.3300 | 0.4543 |
| | | RMSE | 0.2407 | 0.2654 | 0.3020 | 0.3402 | 0.4111 | 0.4099 | 0.43429 | 0.5072 | 0.6734 |
| | $\text{CBiT}_{\text{fused}}$ | MAE | 0.1573 | 0.1734 | 0.1903 | 0.2050 | 0.2578 | 0.2434 | 0.2761 | 0.3466 | 0.4912 |
| | | RMSE | 0.2583 | 0.2850 | 0.3135 | 0.3263 | 0.4044 | 0.3875 | 0.4323 | 0.5267 | 0.7072 |
| | CBiT | MAE | 0.1290 | 0.1517 | 0.1691 | 0.1938 | 0.2154 | 0.2391 | 0.2749 | 0.3284 | 0.4536 |
| | | RMSE | 0.2183 | 0.2732 | 0.2976 | 0.3183 | 0.3575 | 0.3833 | 0.4256 | 0.5022 | 0.6689 |

# F. Module Analysis

## F.1. Past paradigm vs. CTIP: Impact on Temporal Imputation Performance

To verify the effectiveness of the CTIP paradigm in time-series imputation, we adopt the Transformer (Vaswani et al., 2017) as a unified backbone model and conduct comparative experiments between the proposed CTIP and the past paradigm. Under the CTIP setting, historical information is explicitly injected by enabling non-zero history lengths, whereas under the past paradigm, historical conditioning is blocked by setting the history length to zero during training. Experiments are conducted on the ETTh1 dataset under the MCAR missing mechanism and the Joint Supervision training strategy, with missing rates ranging from 10% to 50%. The results are illustrated in Figure 7. As observed, under identical model architectures and training configurations, the CTIP paradigm consistently yields superior imputation performance, validating its effectiveness for time-series imputation.

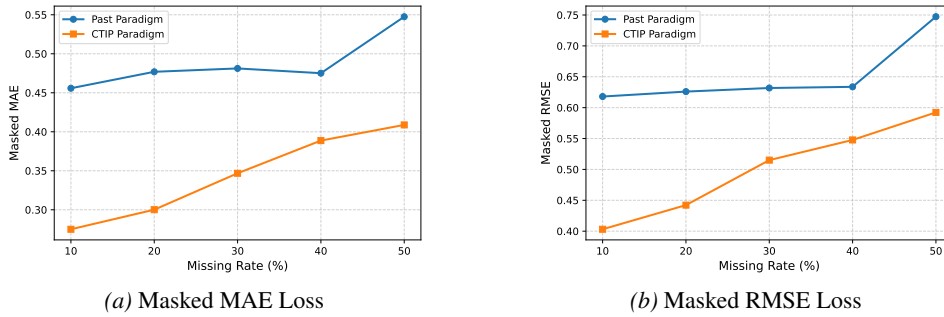

*(a)* Masked MAE Loss         *(b)* Masked RMSE Loss

*Figure 7.* Comparison of the past paradigm and CTIP on ETTh1 under varying missing rates.

