# OpenReview forum: "Rethinking Time-Series Imputation as Conditional Inference along Temporal Evolution"
_ICML.cc/2026/Conference — ICML 2026 regular_

### Official Review · Reviewer_jfbj · 2026-03-09

**Soundness:** 2
**Presentation:** 3
**Significance:** 2
**Originality:** 3
**Overall Recommendation:** 4
**Confidence:** 4

**Summary:**

This paper re-examines the time-series imputation problem from the perspective of temporal state evolution. The authors first propose the Conditional Temporal Inference Paradigm (CTIP), which formulates time-series imputation as conditional inference along the trajectory of temporal evolution. Within this paradigm, CBiT is introduced to perform history-conditioned temporal imputation by employing a history compression mechanism that encodes long-range historical information into a compact latent representation. Furthermore, a partitioned modeling strategy is adopted to explicitly separate historical context from imputation targets while maintaining linear-time computational complexity. Extensive experiments on multiple public benchmarks demonstrate that CBiT significantly improves imputation performance, reducing Masked MAE and Masked RMSE by 27.3% and 18.6%, respectively, across different missing rates.

**Compliance With Llm Reviewing Policy:**

Affirmed.

**Final Justification:**

I appreciate the authors' detailed response to my concerns. Therefore, I have set my score to 4.

**Key Questions For Authors:**

1. The temporal evolution is not clearly reflected in the proposed framework. On the one hand, the authors argue that existing methods for imputation treat time as an indexing mechanism rather than a state-evolving dimension. However, the analysis in Section 3.1 adopts a (Lat, Imp) formulation, which in practice resembles an encoder-style architecture: the latent representation is first constructed and then further processed via cross-attention for imputation. This design appears conceptually similar to prior encoder-based reconstruction approaches. Therefore, the authors should more clearly explain the mathematical meaning of the (Lat, Imp) formulation. As currently presented, the description of temporal evolution suggests a formulation closer to a dynamical system perspective or state-based rather than token sets or hidden states, which is not fully reflected in the model design.

2. On the other hand, according to the definition of the missingness setting in the paper, the model seems to only consider historical observations prior to the missing entries when performing imputation, while potentially overlooking more complex dependency structures. Under this definition, the formulation appears closer to that of a forecasting problem rather than a general imputation setting.

3. It is unclear whether Definition 2.2 implicitly assumes that missing observations occur in contiguous segments. As you say $\mathbf{X}_{\mathrm{imp}}=\{x_t\}_{t=L+1}^{L+\Delta}$, I think you assume that missing observations occur in contiguous segments, which do not always hold in time series imputation task and it is more suitable for forecasting task.

4. The selection of baselines for the model is debatable. The paper does not include the most recent imputation methods as baselines; instead, it compares the proposed approach with relatively recent forecasting methods and older imputation methods. The authors should include comparisons with more up-to-date imputation approaches.

5. In addition, the authors emphasize the advantages of the proposed method for long time series and high-dimensional settings. However, the datasets used in the experiments do not actually reflect such scenarios.

**Limitations:**

yes

**Strengths And Weaknesses:**

**Strengths**:
1. The authors’ analysis effectively identifies the limitations of existing methods and, through reasonable assumptions, reformulates the time-series imputation problem in a principled manner. Furthermore, the paper provides a thorough and accurate analysis of the time-series imputation task.
2. The presentation is clear and easy to understand.
3. The superiority of the proposed method is confirmed in the experiments.

**Weakness**:
Please refer to the question part

---

> ### Author Rebuttal · Authors · 2026-03-30
>
> We sincerely thank you for your valuable comments. We are encouraged by your recognition of the novelty and significance of our work.  Due to space limitations, only partial results are reported here.Below are your concerns and our corresponding responses:
> ### Response to Q1:
> Thank you for this important comment.The current presentation may over-suggest an explicit dynamical-system interpretation. In our framework, temporal evolution does not mean an explicit state-transition equation; rather, it means that imputation is formulated as conditional inference on a temporally unfolded historical trajectory, rather than as static local reconstruction from limited local observations.
>
> Accordingly, the proposed factorization is not intended to redefine temporal evolution as token sets. Instead, it serves as a representation-level design for efficiently instantiating CTIP: $Lat$ summarizes long-range historical context, while $Imp$ represents the target states to be inferred under that context. Thus, the distinction from prior encoder-style reconstruction methods lies not in the computation form itself, but in the conditional formulation. We will clarify this point and better explain the mathematical role of this factorization in the revision.
> ### Response to Q2:
> Thank you for this insightful comment. In our formulation, $X_{{imp}}$ is not a future prediction horizon, but a target inference window containing both observed entries $X_{{imp}}^{(m=0)}$ and missing entries $X_{{imp}}^{(m=1)}$ specified by the mask $M$. The model imputes only the missing entries within this window, rather than predicting the entire segment.
>
> Importantly, the historical segment $X_{hist}$ serves as auxiliary conditioning information to enhance temporal dependency modeling, rather than being the sole source of information. The imputation is jointly conditioned on both $X_{hist}$ and the available observations within $X_{imp}$, allowing the model to capture both local and long-range dependencies. Meanwhile, we observe that incorporating historical context consistently improves performance compared to using only local observations, which validates its complementary role. We will clarify this point in the revision.
> ### Response to Q3:
> Thank you for this important comment. Definition 2.2 does not assume that missing observations in $X_{{imp}}$ are contiguous. In our experiments, missing patterns are imposed on the entire time series $\mathbf{X}$ under a given missing rate and missing mechanism (e.g., MCAR), and missing positions are determined by the mask $\mathbf{M}$. Therefore, missing entries may occur at arbitrary time steps and variables in both $X_{{hist}}$ and $X_{{imp}}$, as they are simply two temporal windows partitioned from the same incomplete sequence and thus share the same missing setting. The target segment $X_{{imp}}$ contains both observed entries $X_{{imp}}^{(m=0)}$ and missing entries $X_{{imp}}^{(m=1)}$, and our goal is to recover $X_{{imp}}^{(m=1)}$ indicated by $\mathbf{M}$, rather than predict a fully missing future segment. We will clarify this more explicitly in the revision.
> ### Response to Q4:
> We further consider, MTSCI(2024)[1], and T1(2026)[2] as additional baselines. Due to space limitations, partial results are shown below, demonstrating that CBiT achieves better performance across different datasets and missing rates.
> |Dataset|Missing|MTSCI MAE|RMSE|T1 MAE|RMSE|CBiT MAE|RMSE|
> |---|---|---|---|---|---|---|---|
> |ETTh1|10%|0.2248|0.3379|0.2463|0.3793|0.1658|0.2849|
> ||30%|0.2570|0.3998|0.2976|0.4415|0.1959|0.3282|
> ||50%|0.3093|0.4879|0.3261|0.5031|0.2217|0.3670|
> |Solar|10%|0.1556|0.2871|0.1101|0.2102|0.0621|0.1412|
> ||30%|0.1515|0.2926|0.1210|0.2281|0.0682|0.1491|
> ||50%|0.1846|0.3789|0.1274|0.2368|0.0856|0.1703|
> ### Response to Q5:
>  In our formulation, “long” mainly refers to long-range temporal dependency rather than only the absolute sequence length. In time-series imputation, missing entries may depend on observations far apart in time, making the problem inherently non-local. Our method is designed for this setting through history-conditioned inference with compressed historical context. In addition, to evaluate scalability on higher-dimensional data, experiments are additionally conducted on the ECL dataset with 321 variables. CBiT still maintains strong performance on this higher-dimensional benchmark. The additional results are shown below.
>
> | Dataset|Missing Rate|XPatch MAE|RMSE|SAITS MAE|RMSE|ImputeFormer MAE |RMSE|CBiT MAE |RMSE |
> |---|---|---|---|---|---|---|---|---|---|
> |ECL|10%|0.2631|0.3705|0.3502|0.5034|0.3159|0.4710|0.2130|0.3309|
> ||30%|0.3337|0.4596|0.3684|0.5203|0.3216 |0.4860|0.1829|0.2825|
> ||50%|0.4078|0.5503|0.3635|0.5178|0.3400|0.5061|0.2108|0.3221|
> References:
>
> [1] Zhou, Jianping, et al. "Mtsci: A conditional diffusion model for multivariate time series consistent imputation."
>
> [2] Park, Dongik, et al. "T1: One-to-One Channel-Head Binding for Multivariate Time-Series Imputation."

---

> > ### Author Rebuttal · Reviewer_jfbj · 2026-04-01
> >
> > My concerns have been adequately addressed and I acknowledge that I have read the author rebuttal. I understand that Area Chairs may flag insufficient reviews during the Reviewer-AC Discussion period and shortly thereafter to address irresponsible, insufficient, or otherwise problematic reviewer conduct. Area Chairs will also be able to flag up during Metareview grossly irresponsible reviewers (including but not limited to neglect of review duties). I understand that my review and my conduct are subject to the ICML Peer Review Ethics (https://icml.cc/Conferences/2026/PeerReviewEthics), and that grossly irresponsible behavior may result in the desk rejection of my co-authored papers.

---

### Official Review · Reviewer_5Gbr · 2026-03-11

**Soundness:** 3
**Presentation:** 3
**Significance:** 3
**Originality:** 3
**Overall Recommendation:** 4
**Confidence:** 4

**Summary:**

The paper studies multivariate time-series imputation and highlights the importance of incorporating historical context during inference. It introduces a new paradigm called Conditional Temporal Inference (CTIP), which models missing values conditioned on both current observations and historical sequences. Based on this idea, the authors propose a model named CBiT that compresses historical information into latent representations to enable efficient long-context inference. Experiments on several benchmark datasets show improved imputation performance compared with existing methods.

**Compliance With Llm Reviewing Policy:**

Affirmed.

**Key Questions For Authors:**

Q1: The paper claims that history compression reduces the complexity from $O((LC)^2)$ to $O(N^2)$. However, the Temporal Latent Writer relies on Perceiver-style cross-attention with cost $O(NLC)$. Could the authors clarify whether the $O(N^2)$ complexity refers only to the latent inference stage rather than the overall module?

Q2: The proposed method relies on compressing historical information into a latent representation of size $N$. How sensitive is the model performance to the choice of $N$ or the amount of historical context? Additional analysis could help understand the robustness of the proposed approach.

Q3: While Section 3 provides a theoretical distinction between CTIP and local reconstruction methods, many Transformer-based baselines (e.g., SAITS, ImputeFormer) also incorporate historical observations through sliding windows. Could the authors clarify what architectural advantage CTIP provides beyond simply concatenating historical tokens in standard self-attention?

**Limitations:**

The paper would benefit from a brief discussion of potential limitations, such as scalability to very high-dimensional time-series data and the sensitivity to historical context length.

**Strengths And Weaknesses:**

Strengths:

S1: The paper addresses an important problem in multivariate time-series imputation and highlights the role of historical context during inference, which is often underutilized in existing approaches.

S2: The proposed framework is technically well motivated and combines temporal modeling with an efficient history compression mechanism to handle long historical sequences.

S3: The overall architecture is relatively clean and modular, separating history compression, cross-block attention, and response aggregation, which improves the interpretability of the design.S4: Extensive experiments across multiple benchmarks demonstrate consistent superiority over strong baselines under varying missing rates and supervision settings.

Weaknesses：

W1: Several important figures are not clearly explained. For instance, Figure 2 is only briefly described as a conceptual comparison, while the definition of Dual-Manifold Cross-Block Attention in Figure 3 remains abstract and Figure 4 is not explicitly referenced in the text.

W2: The Lat-Imp Response Aggregation module introduces multiple components (pooling, fusion, gating), but the motivation is limited. Stronger ablation studies are needed to justify the necessity of these components.

W3: The dimensionality of several datasets (Air, Power, Solar) is not explicitly reported, making it difficult to evaluate scalability with respect to variable dimension.

W4: Although the method claims improved efficiency, the experiments do not report runtime or memory usage, nor analyze how performance and runtime scale with longer historical contexts.

---

> ### Author Rebuttal · Authors · 2026-03-30
>
> We sincerely thank you for your valuable comments. We are encouraged by your recognition of the novelty and significance of our work. Due to the space limit, only partial results are provided below. Below are your concerns and our corresponding responses
>
> ### Response to Weaknesses：
> We thank the reviewer for these helpful comments.
>
> For W1, we thank the reviewer for pointing this out. Figure 2 is intended to provide a conceptual comparison with local reconstruction, while Figures 3 and 4 illustrate the overall architecture and the DMCBA (Dual-Manifold Cross-Block Attention) mechanism, respectively. In the revised version, their roles and connections to the main text will be explained more explicitly, with more explicit descriptions.
>
> For W2, the Lat–Imp Response Aggregation module is introduced based on empirical validation in our experiments and is retained because it consistently improves imputation performance. The results are shown in the table below. In the revision, we will provide a more explicit explanation of the design rationale and include additional ablation results for support.
> | Dataset | MR | w/o Aggregation (MAE) | w/o Aggregation (RMSE) | CBiT (MAE) | CBiT (RMSE) |
> |--------|----|----------------------:|-----------------------:|-----------:|------------:|
> | ETTh1 | 10% | 0.1687 |0.2892 |0.1658|0.2849|
> ||20% | 0.1903 | 0.3042 | 0.1750|0.2956|
>
> For W3,the datasets in the current manuscript are all below 50 variables. To further evaluate scalability on higher-dimensional data, experiments are additionally conducted on the ECL dataset with 321 variables. CBiT still maintains strong performance on this higher-dimensional benchmark. Dataset dimensionalities will be explicitly reported in the revised manuscript.
> | Dataset | Missing Rate | XPatch MAE | XPatch RMSE | SAITS MAE | SAITS RMSE | ImputeFormer MAE | ImputeFormer RMSE |   CBiT MAE |  CBiT RMSE |
> | ------- | -----------: | ---------: | ----------: | --------: | ---------: | ---------------: | ----------------: | ---------: | ---------: |
> |ECL| 10%|     0.2631 |0.3705 |0.3502 |0.5034 |0.3159 | 0.4710 |0.2130|0.3309|
> ||30%|0.3337 |0.4596 | 0.3684 |0.5203 |0.3216 |0.4860 | 0.1829|0.2825|
> ||50%|0.4078 |0.5503 | 0.3635 |0.5178 |0.3400 | 0.5061 |0.2108|0.3221|
>
> For W4, we additionally report runtime and memory usage under the same imputation size. Specifically, CBiT achieves 0.3268s / 34.62MB, compared to 0.4937s / 34.67MB for ImputeFormer and 0.3618s / 20.91MB for SAITS. This shows that CBiT achieves lower runtime with comparable memory consumption, demonstrating its practical efficiency advantage.
>
> We further analyze the effect of different history lengths on ETTh1, as shown below.
> | Dataset | Missing Rate | History Length | MAE | RMSE |
> |--------|-------------:|---------------:|----:|-----:|
> |ETTh1|10% | 24  | 0.1720 | 0.3149 |
> |||48|0.1877 | 0.3220 |
> |||72|0.1957 | 0.3526 |
> |||96|0.2042 | 0.3362 |
> |||120|0.1968 | 0.3312 |
>
>
> ### Response to Questions ：
> For Q1, we thank the reviewer for the helpful comment. We clarify that the $O(N^2)$ term in the current manuscript refers to the latent-space inference stage only, and does not include the history compression step based on Perceiver-style cross-attention, whose complexity is $O(NLC)$. We agree that this point was not described clearly enough in the current version. We will revise the main text to state this more precisely and explicitly.
>
> For Q2,We thank the reviewer for this important question. Since CBiT compresses long-range historical context into a compact latent space, the number of latent tokens \(N\) controls the trade-off between history representation capacity and computational cost. The results are shown below.
> | Dataset | Missing Rate | N=4 |  | N=8 |  | N=16 |  | N=32 |  | N=48 |  |
> |:--|:--:|--:|--:|--:|--:|--:|--:|--:|--:|--:|--:|
> |||MAE | RMSE | MAE | RMSE | MAE | RMSE | MAE | RMSE | MAE | RMSE |
> |ETTh1 | 10% | 0.1950 | 0.3399 | 0.1894 | 0.3249 | 0.1969 | 0.3406 | 0.1872 | 0.3171 | 0.1999 | 0.3357 |
> ||30% | 0.2001 | 0.3581 | 0.2019 | 0.3464 | 0.1972 | 0.3489 | 0.2021 | 0.3511 | 0.1982 | 0.3452|
> ||50% | 0.2654 | 0.4208 | 0.2454 | 0.4016 | 0.2395 | 0.3960 | 0.2330 | 0.3798 | 0.2353 | 0.3833 |
>
> From these results, we observe that performance does change with \(N\), but the variation is overall moderate rather than drastic, suggesting that CBiT is reasonably robust to this hyperparameter. Importantly, the best results are usually achieved by intermediate latent sizes.
>
> For Q3,CTIP goes beyond simply concatenating historical tokens into standard self-attention. It treats historical context and imputation targets as asymmetric conditional components, rather than a homogeneous token set. In CBiT, this is realized by separate history compression and target modeling. We also observe experimentally that removing this distinction and directly concatenating tokens causes clear performance degradation. We will clarify this point more explicitly in the revised manuscript.

---

### Official Review · Reviewer_5Y1Q · 2026-03-12

**Soundness:** 3
**Presentation:** 3
**Significance:** 2
**Originality:** 3
**Overall Recommendation:** 4
**Confidence:** 3

**Summary:**

This paper proposes a CTIP, which reformulates time series imputation from the traditional local missing value reconstruction to a conditional inference process along long-term temporal evolution. The paper also design the CBiT model, which improves imputation accuracy and computational efficiency by compressing historical information and using a dual-manifold attention mechanism.

**Compliance With Llm Reviewing Policy:**

Affirmed.

**Final Justification:**

My initial concerns primarily centered on the rigor of the experiment in the paper. Based on the author’s response, I believe the suggested revisions will adequately address these issues. Therefore, I will give the paper a weak accept score. Since the concerns regarding the rigor of the experiment have been addressed, and taking into account the paper’s overall rigor, originality, significance, and clarity, I still award it a “weak accept” rating.

**Key Questions For Authors:**

The comparison with more recent methods is not sufficient, add more methods based on diffusion models can further enhance its persuasiveness.



The paper lacks enough ablation studies to help the reader better understand the contribution of each component proposed. More detailed experiments are needed to increase the persuasiveness of the paper’s view. The DMCBA(Dual-Manifold Cross-Block Attention) lacks  experiments with independent quadratic attention to compare them in performance and efficiency. The weight λ of the is fixed at 0.5 when use observed loss. The optimal value of this hyperparameter may different across different datasets. Experiments involving system ablation to support the generality of λ would enhance the credibility of the paper’s experimental results. The robustness of CBiT under conditions of long periods of continuous missing data has not been discussed.




The complexity analysis is not very comprehensive.

**Limitations:**

No. Time series interpolation predicts missing values based on past  so the issue of data bias cannot be overlooked.

**Strengths And Weaknesses:**

Advantages:



The writer going to solves a classic question named time series imputation in research area of time series. Although topic is classic, but is significant enough.



The motivation is clear. This paper reformulates time series imputation from the traditional local missing value reconstruction to a conditional inference process along long-term temporal evolution.







The paper rethinks the time series imputation problem, which is somewhat inspiring.







Since long sequence modeling is a core challenge in time series, the proposed history compression design has some practical value.







The experiments are relatively sufficient, and the results show clear performance improvement.



Weaknesses:



The comparison with more recent methods is not sufficient, especially methods based on diffusion models.



The paper lacks enough ablation studies to help the reader better understand the contribution of each component proposed. More detailed experiments are needed to increase the persuasiveness of the paper’s view. The DMCBA(Dual-Manifold Cross-Block Attention) lacks  experiments with independent quadratic attention to compare them in performance and efficiency. The weight λ of the is fixed at 0.5 when use observed loss. The optimal value of this hyperparameter may different across different datasets. Experiments involving system ablation to support the generality of λ would enhance the credibility of the paper’s experimental results. The robustness of CBiT under conditions of long periods of continuous missing data has not been discussed.



The complexity analysis is not very comprehensive.

---

> ### Author Rebuttal · Authors · 2026-03-30
>
> We sincerely thank you for your valuable comments. We are encouraged by your recognition of the novelty and significance of our work. Below are your concerns and our corresponding responses:
>
> ### Response to the comparison with recent diffusion-based methods
>
> We additionally include two diffusion-based baselines, CSDI [1] and MTSCI [2], for further comparison.
>
> | Dataset | Missing Rate | CSDI MAE | CSDI RMSE | MTSCI MAE | MTSCI RMSE | CBiT MAE | CBiT RMSE |
> | :------ | :----------: | -------: | --------: | --------: | ---------: | -------: | --------: |
> | ETTh1   |      10%     |   0.2499 |    0.3728 |    0.2248 |     0.3379 |   0.1658 |    0.2849 |
> | |      30%     |   0.2418 |    0.4008 |    0.2570 |     0.3998 |   0.1959 |    0.3282 |
> | |      50%     |   0.2892 |    0.4675 |    0.3093 |     0.4879 |   0.2217 |    0.3670 |
> | ETTm1|      10%     |   0.1236 |    0.2063 |    0.1282 |     0.2019 |   0.0998 |    0.1835 |
> | |      30%     |   0.1699 |    0.2860 |    0.1460 |     0.2335 |   0.1109 |    0.1929 |
> | |      50%     |   0.1991 |    0.3567 |    0.1781 |     0.2785 |   0.1280 |    0.2275 |
> | Solar | 10%|   0.1150 |    0.2177 |    0.1556 |     0.2871 |   0.0621 |    0.1412 |
> | | 30% |   0.1354 |    0.2575 |    0.1515 |     0.2926 |   0.0682 |    0.1491 |
> | | 50% |   0.2369 |    0.3598 |    0.1846 |     0.3789 |   0.0729 |    0.1572|
>
> After this comparison, we can see that our model still achieves strong performance.
>
> ### Response to the ablation study concern
>
> Due to the space limitation, we are unable to include all ablation results here and therefore only report the studies on DMCBA and $\lambda$.
>
> We compare DMCBA with standard Self-Attention(SA) under the same CBiT framework, where we only replace DMCBA with Self-Attention and keep all other settings unchanged. The results are shown below. We can see that DMCBA performs better in most settings.
>
> | Dataset | Missing Rate | SA MAE | SA RMSE | DMCBA MAE | DMCBA RMSE |
> | :------ | :----------: | -----------------: | ------------------: | --------: | ---------: |
> |ETTh1|10% | 0.2491|0.3952 |0.1658|0.2849|
> || 30%|0.2197 |0.3665 |0.1959 |0.3282 |
> || 50%|0.2714 |0.4293 |0.2217 |0.3670 |
> | Power|10%|  0.0441|0.1100 |0.0336|0.0874|
> ||30%|0.0521 |0.1273 |0.0397|0.1045 |
> ||50%|0.0666 |0.1626 |0.0557|0.1502 |
>
> Second, regarding λ, we agree that the optimal value may vary across datasets. In this work, we use a unified λ to keep the training setting consistent across datasets and baselines for a fair comparison. We also report results under missing-only supervision, where the observed loss is removed, to reflect the model’s own learning ability. In addition, we compare different λ values, and the results are shown below. Since it is difficult to determine the optimal λ for each setting, we use a unified value here. We also observe that the relative effectiveness of different  λ values can vary under different missing rates.
>
> | Dataset | Missing Rate | λ   | UNiTS MAE | UNiTS RMSE | XPatch MAE | XPatch RMSE | CBiT MAE | CBiT RMSE |
> | :------ | :----------: | :-: | --------: | ---------: | ---------: | ----------: | -------: | --------: |
> | ETTh1   |     10%      | 0.1 | 0.3176 | 0.4733 |0.2883 |      0.4261 |   0.2102 |    0.3454 |
> |||0.3 | 0.3370 | 0.5070 |0.2931|0.4291 |0.2344 |0.3770 |
> |||0.5 | 0.3148 | 0.4708 |0.2965|0.4323 |0.1658 |0.2849 |
> |||0.7 | 0.2886 | 0.4403 |0.2971|0.4329 |0.1797 |0.3361 |
> ||| 1.0| 0.3150 | 0.4765 |0.3028|0.4419 |0.1840 |0.3268 |
>
> ### Response to the concern on robustness under extreme missing conditions
>
> We thank the reviewer for pointing this out.In our setting, the history window and the target imputation window are assigned the same missing rate, meaning that high missing rates correspond to a highly challenging incomplete-data scenario. As shown in Table 7 and Table 8, CBiT still maintains competitive performance even at the extreme 90\% missing rate, suggesting its robustness under severely incomplete conditions. This aspect will also be presented more explicitly in the revised version.
>
> ### Response to the concern on complexity analysis
>
> We thank the reviewer for raising this point. In the previous version, the complexity discussion mainly focused on the compressed inference stage, while the cost of the compression step was not explicitly included. More specifically, for historical-context processing, the compression step has complexity $O(NLC)$, and the subsequent inference in the compressed token space has complexity $O(N^2)$. Accordingly, the overall complexity becomes $O(NLC+N^2)$, which remains substantially lower than the original $O((LC)^2)$ cost of applying full self-attention over the history space. The revised version will make this point more explicit.
>
> References:
>
> [1] Tashiro, Yusuke, et al. "Csdi: Conditional score-based diffusion models for probabilistic time series imputation."
>
> [2] Zhou, Jianping, et al. "Mtsci: A conditional diffusion model for multivariate time series consistent imputation."

---

> > ### Author Rebuttal · Reviewer_5Y1Q · 2026-04-01
> >
> > My concerns have been addressed.

---

### Official Review · Reviewer_ShNc · 2026-03-12

**Soundness:** 3
**Presentation:** 3
**Significance:** 3
**Originality:** 4
**Overall Recommendation:** 4
**Confidence:** 3

**Summary:**

This paper proposes the Conditional Temporal Inference Paradigm (CTIP), which reframes time-series imputation from local conditional reconstruction to conditional inference along temporal evolution. The authors introduce CBiT, a Transformer-based imputation model that compresses long-range historical context into latent representations and uses a partitioned cross-block design to distinguish historical context from imputation targets in a computationally scalable manner. Experiments on multiple benchmarks show that CBiT achieves consistently strong imputation performance across different missing rates and supervision settings, with additional results suggesting robustness under more extreme missingness. Overall, the paper argues that explicitly conditioning on long-term temporal evolution can improve time-series imputation beyond reconstruction-oriented baselines.

**Compliance With Llm Reviewing Policy:**

Affirmed.

**Final Justification:**

The authors have adequately addressed my concerns in the rebuttal. Therefore, I maintain my score and recommend weak acceptance.

**Key Questions For Authors:**

1. Can the authors provide results for a more challenging setting where the history window $X_{hist}$ also contains missing values, potentially at varying missing rates?
2. How sensitive is performance to the number of latent tokens $N$, and how is $N$ chosen across datasets with different temporal characteristics?

**Limitations:**

yes

**Strengths And Weaknesses:**

Soundness:
This paper is reasonably sound. Expected-risk analysis gives a plausible justification for using long-range historical context in imputation, and the empirical evaluation is broad across eight benchmarks, various missing rates, and supervision settings. However, the theory mainly motivates the paradigm rather than the specific model architecture, and the experiments do not fully examine robustness when the historical context itself is heavily missing or corrupted.

Presentation:
The paper is clear and well-organized. Figure 2 effectively conveys the conceptual motivation behind the CTIP paradigm, while Figure 3 provides a clear exposition of the CBiT architecture. However, the empirical justification for certain design choices, such as the number of latent tokens $N$, is limited and would benefit from a rigorous sensitivity analysis.

Significance:
The paper presents a significant conceptual and practical contribution by shifting the imputation paradigm toward conditional temporal inference. The proposed CBiT model demonstrates a compelling performance-efficiency trade-off, particularly evidenced by its robustness in extreme missingness scenarios.

Originality:
The paper has solid originality. Its main novelty lies in reframing imputation as conditional inference along temporal evolution, rather than local reconstruction. Although some architectural ingredients, such as latent compression via cross-attention, are related to prior approaches, the partitioned modeling of historical context and imputation targets is a distinct and meaningful task-specific contribution.

---

> ### Author Rebuttal · Authors · 2026-03-29
>
> We sincerely thank you for your valuable comments. We are encouraged by your recognition of the novelty and significance of our work. Below are your concerns and our corresponding responses:
>
> ### (1) Robustness when the history window  $X_{hist}$ is also missing.
>
> In our main experiments, the history window $X_{hist}$  and the target imputation window $X_{imp}$ are assigned the same missing rate. Therefore, our standard setting already considers a more realistic setting where the historical context is itself incomplete, rather than assuming a fully clean history.
> To further answer the reviewer’s question, we conducted an additional stress test in which we injected extra missingness into the history window while keeping the target missing rate fixed. The results on ETTh1 are:
> | Dataset | Missing Rate | Extra Rate=30% |  | Extra Rate=50% |  | Extra Rate=70% |  | No Injection |  |
> |:-------:|:------------:|:-------------:|:--:|:-------------:|:--:|:-------------:|:--:|:------------:|:--:|
> |         |              | MAE | RMSE | MAE | RMSE | MAE | RMSE | MAE | RMSE |
> | ETTh1   | 10%          | 0.1884 | 0.3470 | 0.1835 | 0.3437 | 0.1934 | 0.3537 | **0.1658** | **0.2849** |
> |         | 20%          | 0.2170 | 0.3915 | 0.2137 | 0.3762 | 0.2061 | 0.3711 | **0.1750** | **0.2956** |
>
> We observe that injecting additional missing values leads to performance degradation. However, under the overall historical trend, the model still maintains a reasonable level of performance.
>
> ### (2) Sensitivity to the number of latent tokens \(N\).
>
> We thank the reviewer for this important question. Since CBiT compresses long-range historical context into a compact latent space, the number of latent tokens N controls the trade-off between history representation capacity and computational cost. To study this, we conducted a sensitivity analysis on ETTh1 and ETTh2 with N = {4, 8, 16, 32, 48}.
>
> | Dataset | Missing Rate | N=4 |  | N=8 |  | N=16 |  | N=32 |  | N=48 |  |
> |:--|:--:|--:|--:|--:|--:|--:|--:|--:|--:|--:|--:|
> |  |  | MAE | RMSE | MAE | RMSE | MAE | RMSE | MAE | RMSE | MAE | RMSE |
> | ETTh1 | 10% | 0.1950 | 0.3399 | 0.1894 | 0.3249 | 0.1969 | 0.3406 | 0.1872 | 0.3171 | 0.1999 | 0.3357 |
> |  | 20% | 0.1850 | 0.3340 | 0.1803 | 0.3233 | 0.1742 | 0.3244 | 0.2082 | 0.3549 | 0.1812 | 0.3206 |
> |  | 30% | 0.2001 | 0.3581 | 0.2019 | 0.3464 | 0.1972 | 0.3489 | 0.2021 | 0.3511 | 0.1982 | 0.3452|
> |  | 40% | 0.2157 | 0.3719 | 0.2115 | 0.3568 | 0.2149 | 0.3719 | 0.2152 | 0.3697 | 0.2187 | 0.3732 |
> |  | 50% | 0.2654 | 0.4208 | 0.2454 | 0.4016 | 0.2395 | 0.3960 | 0.2330 | 0.3798 | 0.2353 | 0.3833 |
> | ETTh2 | 10% | 0.1366 | 0.1940 | 0.1438 | 0.2042 | 0.1597 | 0.2267 | 0.1878 | 0.2668 | 0.2356 | 0.3266 |
> |  | 20% | 0.1235 | 0.1775 | 0.1171 | 0.1674 | 0.1180 | 0.1728 | 0.1451 | 0.1993 | 0.1230 | 0.1768 |
> |  | 30% | 0.1279 | 0.1825 | 0.1287 | 0.1896 | 0.1223 | 0.1769 | 0.1723 | 0.2382 | 0.1845 | 0.2534 |
> |  | 40% | 0.1484 | 0.2148 | 0.1386 | 0.2002 | 0.1903 | 0.2657 | 0.1477 | 0.2131 | 0.1457 | 0.2097 |
> |  | 50% | 0.1554 | 0.2215 | 0.1597 | 0.2284 | 0.1640 | 0.2332 | 0.1541 | 0.2203 | 0.1588 | 0.2311 |
>
> ### Computational Cost
>
> | N      | 4      | 8      | 16     | 32     | 48     |
> |--------|--------|--------|--------|--------|--------|
> | GFLOPs | 0.0587 | 0.0606 | 0.0643 | 0.0718 | 0.0793 |
>
> From these results, we observe that performance does change with N, but the variation is overall moderate rather than drastic, suggesting that CBiT is reasonably robust to this hyperparameter. Importantly, the best results are usually achieved by intermediate latent sizes (e.g., 8, 16, or 32), rather than by the smallest or largest settings.
>
> This indicates that the role of latent tokens is not simply to maximize model capacity, but to achieve an appropriate balance between temporal information preservation and compact history compression. When N is too small, the latent bottleneck may be overly restrictive and fail to preserve sufficient long-range temporal dynamics. In contrast, when N becomes too large, the model gains more representational freedom, but the benefit of compact summarization becomes weaker, while computational cost increases.
>
> We also observe a clear efficiency trend: as N increases from 4 to 48, the parameter count remains essentially unchanged, while GFLOPs increase gradually from 0.0587 to 0.0793. This suggests that N mainly affects computational overhead, and we find that N has little impact on model size.Therefore, considering both performance and efficiency, a  moderate latent size provides the best trade-off. Based on this observation, we use a middle-range default setting in the main paper, and we will include this sensitivity analysis in the revised version to better justify the choice of N.

---

> > ### Author Rebuttal · Reviewer_ShNc · 2026-04-03
> >
> > Thank you for the detailed rebuttal. The authors have adequately addressed my main questions, especially by providing additional experiments on the case where the history window is also incomplete and by adding a sensitivity analysis for the number of latent tokens N. These additions make the empirical picture clearer and strengthen the justification for the design choices. My overall assessment remains positive and unchanged, and I keep my score at 4.

---

### Decision · Program_Chairs · 2026-04-30

**Decision:**

Accept (regular)

**Comment:**

This paper proposes the Conditional Temporal Inference Paradigm (CTIP), which reframes time series imputation as conditional inference along temporal evolution rather than local reconstruction, and introduces CBiT, a model that compresses long-range history into compact latent representations, improving imputation accuracy and computational efficiency.

All reviewers acknowledged the clear motivation of the work, the originality of the paradigm shift, and the strong empirical results across multiple benchmarks. Key concerns included insufficient comparison to more recent baselines, limited ablation studies, method robustness, unclear complexity/runtime analysis, and whether the formulation truly reflects temporal evolution rather than an encoder-based reconstruction.

The rebuttal was thorough, providing further comparisons to recent methods, with additional ablation results, robustness and sensitivity analyses, computational cost report, and clarifications on the reviewers' questions. Given the unanimous positive assessment and the adequate rebuttal, while three reviewers confirmed that their concerns were fully resolved, I recommend to accept the paper as a poster. I encourage the authors to integrate the additional experimental results and promised clarifications into the camera-ready version, and better explain why the formulation constitutes imputation rather than forecasting.